# Identification of pyrogallol as a warhead in design of covalent inhibitors for the SARS-CoV-2 3CL protease

Haixia Su [1,2,9], Sheng Yao [2,3,9], Wenfeng Zhao [1,9], Yumin Zhang [4,9], Jia Liu [2,3,9], Qiang Shao [1,9], Qingxing Wang [2,4], Minjun Li[5], Hang Xie[1], Weijuan Shang[4], Changqiang Ke[3], Lu Feng[3], Xiangrui Jiang[1,2], Jingshan Shen [1,2], Gengfu Xiao[2,4], Hualiang Jiang [1,2,6,7], Leike Zhang [2,4,10✉], Yang Ye [2,3,8,10✉] & Yechun Xu [1,2,7,10✉]

The ongoing pandemic of coronavirus disease 2019 (COVID-19) caused by severe acute respiratory syndrome coronavirus 2 (SARS-CoV-2) urgently needs an effective cure. 3CL protease (3CL[pro]) is a highly conserved cysteine proteinase that is indispensable for coronavirus replication, providing an attractive target for developing broad-spectrum antiviral drugs. Here we describe the discovery of myricetin, a flavonoid found in many food sources, as a non-peptidomimetic and covalent inhibitor of the SARS-CoV-2 3CL[pro]. Crystal structures of the protease bound with myricetin and its derivatives unexpectedly revealed that the pyrogallol group worked as an electrophile to covalently modify the catalytic cysteine. Kinetic and selectivity characterization together with theoretical calculations comprehensively illustrated the covalent binding mechanism of myricetin with the protease and demonstrated that the pyrogallol can serve as an electrophile warhead. Structure-based optimization of myricetin led to the discovery of derivatives with good antiviral activity and the potential of oral administration. These results provide detailed mechanistic insights into the covalent mode of action by pyrogallol-containing natural products and a template for design of non-peptidomimetic covalent inhibitors against 3CL[pro]s, highlighting the potential of pyrogallol as an alternative warhead in design of targeted covalent ligands.

[1] CAS Key Laboratory of Receptor Research, and Drug Discovery and Design Center, Shanghai Institute of Materia Medica, Chinese Academy of Sciences, Shanghai, China. [2] University of Chinese Academy of Sciences, Beijing, China. [3] State Key Laboratory of Drug Research, and Natural Products Chemistry Department, Shanghai Institute of Materia Medica, Chinese Academy of Sciences, Shanghai, China. [4] State Key Laboratory of Virology, Wuhan Institute of Virology, Center for Biosafety Mega-Science, Chinese Academy of Sciences, Wuhan, China. [5] Shanghai Synchrotron Radiation Facility, Shanghai Advanced Research Institute, Chinese Academy of Sciences, Shanghai, China. [6] Shanghai Institute for Advanced Immunochemical Studies and School of Life Science and Technology, ShanghaiTech University, Shanghai, China. [7] School of Pharmaceutical Science and Technology, Hangzhou Institute for Advanced Study, University of Chinese Academy of Sciences, Hangzhou, China. [8] School of Life Science and Technology, ShanghaiTech University, Shanghai, China. [9] These authors contributed equally: Haixia Su, Sheng Yao, Wenfeng Zhao, Yumin Zhang, Jia Liu, Qiang Shao. [10] These authors jointly supervised this work: Leike Zhang, Yang Ye, Yechun Xu. ✉email: zhangleike@wh.iov.cn; yye@simm.ac.cn; ycxu@simm.ac.cn

Three highly pathogenic coronaviruses (CoVs), including severe acute respiratory syndrome coronavirus 2 (SARS-CoV-2), SARS-CoV, and Middle East respiratory syndrome coronavirus (MERS-CoV), lead to three epidemics which pose tremendous threats to public health and economics. In particular, the ongoing SARS-CoV-2 pandemic (referred to as coronavirus disease 2019, COVID-19) has caused over 72 million infections and over 1.6 million deaths worldwide, and the numbers are still increasing[1]. As a member of the genus β-coronavirus, SARS-CoV-2 is closely related to many bat coronaviruses and SARS-CoV[2], and it has high human-to-human transmissibility and causes significant mortality in older patients with other co-morbidities[3,4]. To date, remdesivir, an inhibitor of RNA-dependent RNA polymerase (RdRp), is the only drug approved by the FDA for the treatment of COVID-19 in the USA. Therefore, there is an enormous unmet need for the development of antiviral drugs to treat the diseases caused by these pathogenic CoVs.

SARS-CoV-2, SARS-CoV, and MERS-CoV all belong to the genus β-coronavirus and are three of seven known CoVs that cause human diseases. The positive-sense single-stranded RNA genome of these enveloped viruses is translated by host ribosomes into two polyproteins, pp1a and pp1ab. The cleavage of the polyproteins by two cysteine proteases, a chymotrypsin-like protease called 3C-like protease (3CL$^{pro}$) and a papain-like protease (PL$^{pro}$), generates mature non-structural proteins such as RNA-dependent RNA polymerase (RdRp) and helicase, which are essential for the completion of the viral life cycle[5]. There are 11 cleavage sites for 3CL$^{pro}$ in the polyproteins, therefore, 3CL$^{pro}$ is also referred to as the main protease (M$^{pro}$). The substrate specificity of 3CL$^{pro}$ is featured by the efficient cleavage in the peptides including (Leu, Phe, Met, Val)-Gln↓(Ser, Ala, Gly) sequences (the cleavage site is indicated by ↓), and a remarkably high degree of conservation of the substrate-binding sites, particularly for the crucial S1/S2 subsites, has been well-documented[6–8]. The vital role in processing the polyproteins and the highly conserved substrate specificity of 3CL$^{pro}$ make it an attractive target for the development of broad-spectrum antiviral drugs. In general, substrate analogs or mimetics attached with a chemical warhead targeting the catalytic cysteine were designed as peptidomimetic inhibitors of 3CL$^{pro}$ with a covalent mechanism of action[6], none has yet progressed into clinical trials[9–11]. Discovery of more drug-like 3CL$^{pro}$ inhibitors with diverse chemical structures is crucial to speed up the drug development against the highly pathogenic CoVs as 3CL$^{pro}$ is one of the best-characterized drug targets among CoVs[12–15].

Considering the therapeutic benefits, including the high potency, an extended duration of action, a reduced risk for the development of drug resistance, and binding to otherwise "intractable" targets, covalent ligands are of great interest as therapeutic drugs[16–18]. Although many historical covalent ligands were discovered by serendipity, targeted covalent ligand design has experienced a resurgence during the past two decades and it has emerged as a powerful approach to drug discovery[16,19]. Targeting the nucleophile of a specific cysteine or serine residue of enzymes with electrophilic reactive groups, the so-called warheads, is the predominant strategy in targeted covalent inhibitor development[20–22]. For example, boceprevir and telaprevir, two drugs approved by the FDA for the treatment of hepatitis C virus (HCV) infection, both utilize a warhead of ketoamide to covalently react with the catalytic serine of the HCV NS3 protease[23,24]. Noteworthy, there has been a very high interest in characterization of alternative warheads to meet a large variety of requirements in medicinal chemistry and chemical biology, though cysteine/serine-targeted Michael acceptors such as acrylamides and other α,β-unsaturated carbonyls are the predominant warheads in the realm of current covalent drug development.

We have reported previously that baicalein, a natural flavonoid isolated from *Scutellaria baicalensis* Georgi, is a non-covalent inhibitor of SARS-CoV-2 3CL$^{pro}$ with a high ligand binding efficiency[25]. Moreover, a crystal structure of the SARS-CoV-2 3CL$^{pro}$ in complex with baicalein revealed that it utilized a unique binding mode to reversibly inhibit the proteolytic activity of the protease. Inspired by this finding, a series of flavonoids were tested using an enzymatic assay in the present study. As a result, myricetin showed good inhibitory activity against the protease. However, the crystal structure of the SARS-CoV-2 3CL$^{pro}$ bound with myricetin reveals an unexpected covalent binding mode that the pyrogallol moiety of myricetin covalently links to the catalytic cysteine. This discovery not only establishes the molecular mechanism of action of myricetin, but also illuminates the pyrogallol as a warhead suited for engaging the catalytic cysteine of 3CL$^{pro}$. While the intrinsic oxidation reactivity of pyrogallol would normally preclude its use as hit/lead compounds, we demonstrate that it could serve as a good starting point for the development of cysteine-targeted covalent ligand. Insights from our mechanistic studies have led us to rationally design myricetin derivatives as well as prodrugs with improved antiviral activities.

## Results

**Inhibition of the enzymatic activity of SARS-CoV-2 3CL$^{pro}$ and the replication of SARS-CoV-2 in cells by myricetin.** As previously reported, a fluorescence resonance energy transfer (FRET) protease assay was applied to measure the proteolytic activity of the recombinant SARS-CoV-2 3CL$^{pro}$ on a fluorescently labeled substrate, MCA-AVLQSGFR-Lys(Dnp)-Lys-NH$_2$[25]. This FRET-based protease assay was utilized to measure the inhibitory activities of 19 flavonoids against the SARS-CoV-2 3CL$^{pro}$. At a concentration of 10 μM, both myricetin (3,5,7,3′,4′,5′-hexahydroxyflavone, Supplementary Fig. 1a) and dihydromyricetin displayed >90% inhibition against the protease, while the inhibition by other compounds was relatively low (Supplementary Table 1). The half-maximal inhibitory concentration (IC$_{50}$) of myricetin and dihydromyricetin was 0.63 and 1.14 μM, respectively (Fig. 1a, b and Supplementary Table 2). The inhibitory activity of myricetin against the protease is even better than that of baicalein (IC$_{50}$: 0.94 μM). Accordingly, myricetin and dihydromyricetin, two natural flavonoids found in many foods, are identified as inhibitors of the SARS-CoV-2 3CL$^{pro}$ with sub-micromolar or micromolar potency.

We further evaluated the antiviral efficacy of myricetin against SARS-CoV-2 in Vero E6 cells. The cytotoxicity of myricetin in the cells was first determined by the CCK8 assay, and the resulting half-maximal cytotoxic concentration (CC$_{50}$) of the compound was over 200 μM, demonstrating a very low cytotoxicity of the compound (Supplementary Fig. 2). Subsequently, the Vero E6 cells were infected with SARS-CoV-2 at a multiplicity of infection (MOI) of 0.01 in the presence of different concentrations of myricetin. The antiviral efficacy was evaluated by quantification of viral copy numbers in the cell supernatant via quantitative real-time RT-PCR (qRT-PCR). As shown in Fig. 1c, myricetin showed dose-dependent inhibition on the replication of SARS-CoV-2, and the resulting half-maximal effective concentration (EC$_{50}$) was 8.00 μM. As a positive control, remdesivir inhibited the SARS-CoV-2 replication in Vero E6 cells with an EC$_{50}$ value of 3.68 μM. The resulting selectivity index (SI) value is >25 for myricetin. Therefore, the cell-based antiviral experiment demonstrates that myricetin is able to inhibit the viral replication. The EC$_{50}$ of dihydromyricetin was also determined with a value of 13.56 μM (Fig. 1c and Supplementary Table 2).

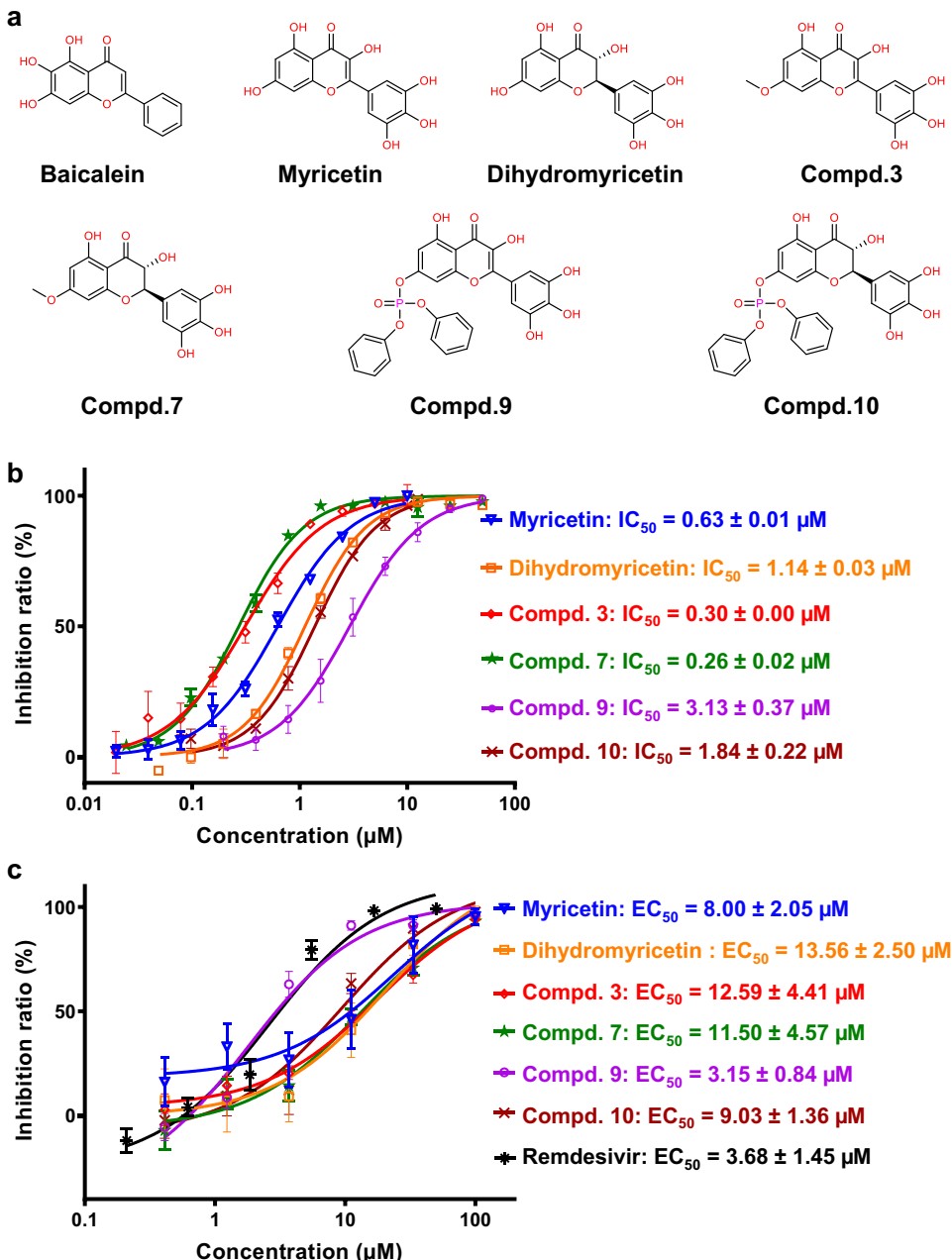

**Fig. 1 Inhibition of the enzymatic activity of the SARS-CoV-2 3CL^pro and the replication of SARS-CoV-2 in cells by myricetin and its derivatives.** **a** Chemical structures of baicalein, myricetin, dihydromyricetin, and compounds **3**, **7**, **9**, and **10**. **b** Representative inhibition profiles for myricetin (blue), dihydromyricetin (orange), **3** (red), **7** (green), **9** (purple), and **10** (dark red) against the SARS-CoV-2 3CL^pro. Error bars represent mean ± SD of three independent experiments. **c** Inhibition profiles of myricetin (blue), dihydromyricetin (orange), **3** (red), **7** (green), **9** (purple), **10** (dark red), and remdesivir (black) against the replication of SARS-CoV-2 in Vero E6 cells. Error bars represent mean ± SD of six independent experiments.

**Crystal structure of the SARS-CoV-2 3CL^pro covalently bound with myricetin**. To understand the binding mode of these inhibitors with the protease, a crystal structure of the SARS-CoV-2 3CL^pro in complex with myricetin was determined at a resolution of 2.1 Å (Supplementary Table 3). Myricetin binds at the catalytic site within the extended substrate-binding pocket of the protease which has a catalytic Cys145-His41 dyad. Rather unexpectedly, continuous electron density was clearly shown between Cys145 and myricetin (Fig. 2a), allowing us to place an exact covalent bond between the sulfur atom of Cys145 and the C6′ atom of the pyrogallol group. In addition to this covalent binding interaction, several hydrogen-bonds (H-bonds) were formed between two hydroxyl groups of the pyrogallol group

and the main chains of Gly143/Ser144/Cys145/Thr26. The chromone moiety of myricetin established H-bonds with the side chain of Glu189 as well as a buried water molecule which simultaneously contacted with His164/His41/Asp187 (Fig. 2a). In addition, it also formed π–π stacking interactions with the side chain of His41. Accordingly, myricetin is perfectly engaged with the catalytic site by making both covalent bonding and non-covalent interactions with the surrounding residues. The crystal structure of the complex thereby provides the unexpected structural insight into the covalent recognition of myricetin by the SARS-CoV-2 3CL^pro, and reveals that the pyrogallol group of myricetin serves as an electrophile to react with the nucleophile of Cys145.

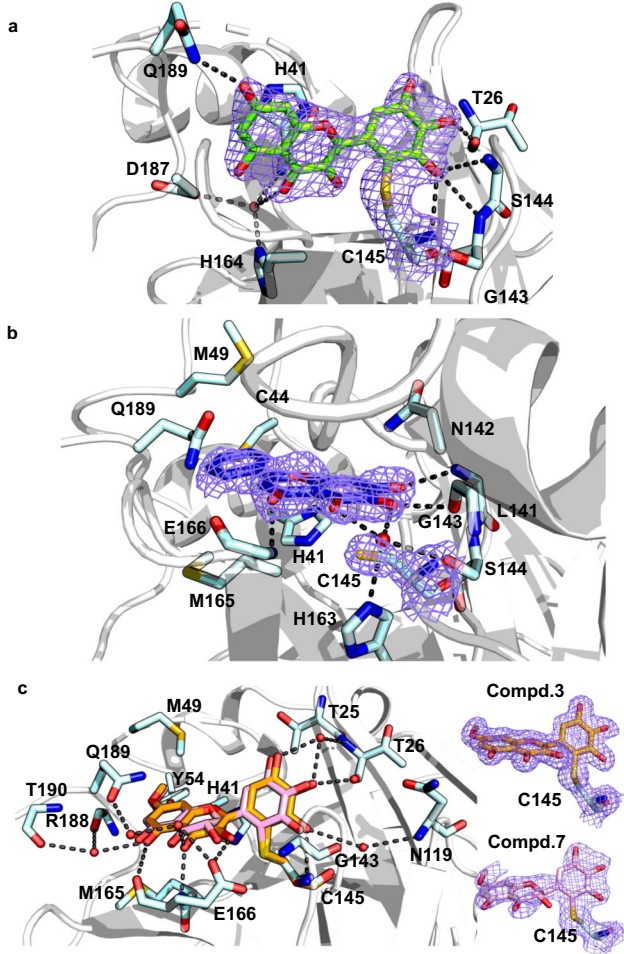

**Fig. 2 Crystal structures of the SARS-CoV-2 3CL^pro in complex with inhibitors.** Binding modes of myricetin (**a**), baicalein (**b**), and compounds **3** and **7** (**c**) with the SARS-CoV-2 3CL^pro. The protease is shown in gray cartoon, myricetin in green sticks, baicalein in blue purple sticks, compound **3** in orange sticks, compound **7** in light pink sticks, and the surrounding residues in palecyan sticks. H-bonds are represented by black dashed lines. 2Fo-Fc density maps are shown in slate for myricetin, baicalein, **3**, and **7** contoured at 1.2σ.

Despite the fact that myricetin and baicalein are inhibitors of the SARS-CoV-2 3CL^pro and both of them possess a flavonoid scaffold as well as a pyrogallol group, the mode of action and the structural determinants associated with their binding with the protease are quite different (Fig. 2a, b). Baicalein is a non-covalent inhibitor of the protease while myricetin establishes a covalent bond with the catalytic Cys145. The orientation of myricetin at the binding site is different from that of baicalein, resulting in distinct ligand-protein interaction patterns. When compared to myricetin, baicalein forms more H-bonding and hydrophobic interactions with the residues. Notably, the pyrogallol group of baicalein forms multiple H-bonds with main chains of Leu141/Gly143 as well as the side chain of Ser144, fixing the conformation of the oxyanion loop (residues 138–145) which serves to stabilize the tetrahedral transition state of the proteolytic reaction. Instead, the pyrogallol group of myricetin acts as an electrophile to covalently bind to Cys145. In addition, in the complex of SARS-CoV-2 3CL^pro with myricetin, the side chain of His41 adopted an orientation opposite to its conformations in most reported crystal structures of the SARS-CoV-2 3CL^pro, including the baicalein-bound one (Supplementary Fig. 1b).

Nevertheless, the side chain of His41 always forms π–π stacking interactions with the chromone region of baicalein or myricetin, demonstrating a pivotal role of His41 in binding with the flavonoid scaffold inhibitors.

Besides the crystal structure determination, the recombinant SARS-CoV-2 3CL^pro incubation with myricetin or DMSO was further analyzed by mass spectrometry. As shown in Fig. 3a, myricetin covalently labeled the protease as indicated by an increase of 316 Da in molecular weight, indicating the modification of the SARS-CoV-2 3CL^pro by myricetin. The crystal structure of the SARS-CoV-2 3CL^pro bound with myricetin and the intact mass spectrometry together reveal that the catalytic Cys145 of the protease was covalently modified by myricetin (Fig. 3b).

**Kinetics characterization of myricetin binding with the SARS-CoV-2 3CL^pro.** In general, selective covalent ligand binding involves a two-step process: an initial reversible binding event followed by formation of the covalent bond, which is best characterized by the binding affinity ($K_i$) and the first-order rate constant of covalent modification ($k_{inact}$), respectively. The resulting second-order rate constant ($k_{inact}/K_i$) provides a preferred measure over the IC$_{50}$ to describe the potency of a covalent inhibitor against a target[21,26]. To obtain the $k_{inact}/K_i$ of myricetin binding with the SARS-CoV-2 3CL^pro, five different concentrations, ranging from 2.5 to 40 μM, of myricetin were incubated with the protease at a final concentration of 100 nM for 250 s and activities of the protease in each reaction were measured at indicated time (Fig. 3c). For each concentration of myricetin, the protease activity was plotted against incubation time to generate the $k_{obs}$ value. The relationship between myricetin concentrations and $k_{obs}$ shown in Fig. 3d resulted in a $k_{inact}$ of 0.011 s$^{-1}$, a $K_i$ of 15.73 μM and a $k_{inact}/K_i$ of 701.88 M$^{-1}$s$^{-1}$. The non-covalent binding affinity ($K_i$) of myricetin with the SARS-CoV-2 3CL^pro is 15.73 μM, suggesting that myricetin binds selectively to the substrate-binding pocket of the protease, which provides a basis for driving the covalent bond formed between myricetin and Cys145. The measured $k_{inact}$ is 0.011 s$^{-1}$, indicating that myricetin could quickly react with the Cys145. Overall, considering the small size of myricetin, it is an efficient covalent binder of the SARS-CoV-2 3CL^pro with a $k_{inact}/K_i$ of 701.88 M$^{-1}$s$^{-1}$.

The acquirement of high selectivity for a covalent inhibitor to reduce off-target reactions requires that the intrinsic reactivity of the electrophilic warhead on the inhibitor should be low. The half-life of the electrophile to react with glutathione (GSH$t_{1/2}$) is a useful assay for measuring the intrinsic reactivity of cysteine-targeted warheads, and for providing information about the electrophilicity and liability toward forming reactive intermediates[24,27,28]. To determine the rates of reaction with GSH, a range of concentrations of myricetin, baicalein or N-phenylacrylamide (a positive control) were incubated with GSH, and the remaining compounds at varying time was determined by liquid chromatography-tandem mass spectrometry (LC-MS). The GSH$t_{1/2}$ of myricetin and N-phenylacrylamide is 497 and 34 min (Fig. 3e, f and Supplementary Figs. 3 and 4), respectively, while no adduct of GSH with baicalein was detected after incubation for 24 h. This result demonstrates that the pyrogallol of myricetin as a warhead shows low reactivity toward GSH, suggesting the low probability of nonspecific binding to cysteine. The pyrogallol of baicalein even has no detectable reactivity with GSH, which is in line with the crystal structure of SARS-CoV-2 3CL^pro reversibly bound with baicalein.

**Selectivity of myricetin toward the SARS-CoV-2 3CL^pro in cell cultures.** To assess whether myricetin maintains high selectivity against extensive cysteine-containing proteins in a cellular

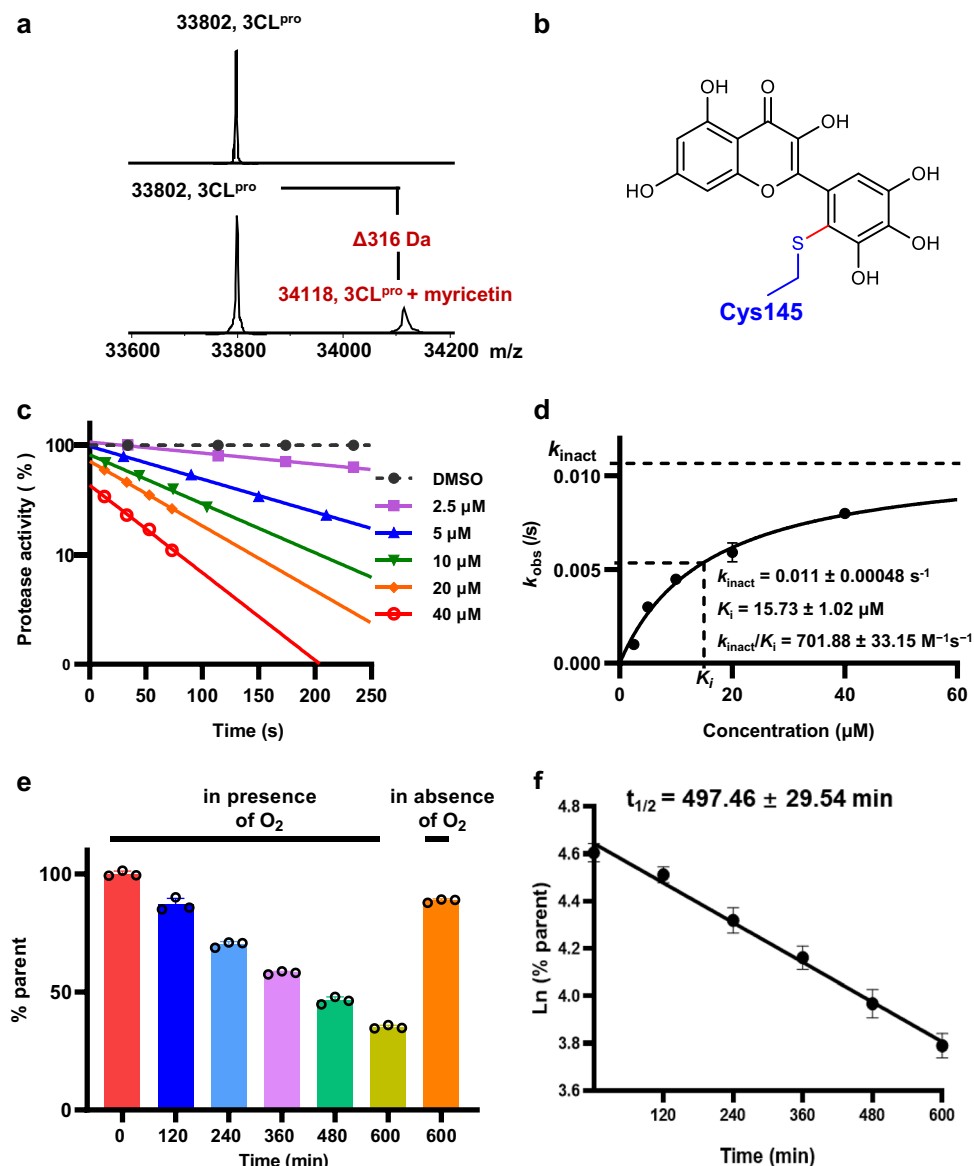

**Fig. 3 Characterization of myricetin binding with the SARS-CoV-2 3CLpro and half-life determination of myricetin reacting with GSH. a** Mass spectrometry analysis for the SARS-CoV-2 3CLpro treated with DMSO or myricetin. Three independent experiments were performed. **b** Proposed reaction adduct of the SARS-CoV-2 3CLpro with myricetin. **c** The SARS-CoV-2 3CLpro (at a final concentration of 100 nM) was incubated with five different concentrations of myricetin (2.5, 5, 10, 20, and 40 μM), respectively. For each concentration, the protease activity at different time was measured by the FRET-based protease assay and plotted against the incubation time to obtain the $k_{obs}$ value (an absolute value of the slope of each linear curve). **d** The resulting $k_{obs}$ values were plotted versus inhibitor concentrations to generate the $k_{inact}$ and $K_i$ values of myricetin binding with the SARS-CoV-2 3CLpro. **e** Myricetin (at a final concentration of 400 μM) was incubated with 10 mM GSH for 0, 120, 240, 360, 480, and 600 min, in the presence or absence of oxygen, respectively. The remaining myricetin was determined by LC-MS. **f** Ln (the percentage of the remaining myricetin) was plotted against incubation time to generate the half-life time of myricetin reacting with GSH. Error bars represent mean ± SD of three independent experiments in Fig. **c–f**.

environment, the specificity of myricetin was examined by an activity-based protein profiling (ABPP) method with HEK293T cells (Supplementary Fig. 5). The gel-based competitive ABPP analysis was performed using fluorescently labeled maleimide as a probe, which could easily conjugate to a thiol group on a protein without selectivity. The recombinant SARS-CoV-2 3CLpro, HEK293T lysate, and the mixture of them were incubated with three different concentrations (0.5, 2, and 10 μM) of myricetin or vehicle, respectively. Then the fluorescent probe was added to label any remaining unreacted cysteine residues followed by an in-gel fluorescence scanning. As shown in Supplementary Fig. 5, myricetin at a concentration of 10 μM was able to completely block the fluorescent labeling of the recombinant SARS-CoV-2 3CLpro by the probe at a concentration of 1 μM. Moreover, incubation of 10 μM myricetin with the mixture of the HEK293T lysate and the recombinant SARS-CoV-2 3CLpro only significantly reduced the fluorescent labeling of the protease, implying the good selectivity of myricetin toward the SARS-CoV-2 3CLpro over the soluble cysteinome in cells.

In addition, we measured the inhibitory activity of myricetin against the SARS-CoV 3CLpro, the SARS-CoV-2 PLpro, and bovine chymotrypsin, to investigate the selectivity of this flavonoid toward the SARS-CoV-2 3CLpro. The results showed that myricetin displayed a comparable inhibitory activity toward the SARS-CoV 3CLpro as it did for the SARS-CoV-2 3CLpro, with

an IC$_{50}$ value of 0.74 ± 0.08 μM. However, the inhibitory activity of myricetin against the SARS-CoV-2 PL$^{pro}$ (IC$_{50}$: 159.10 ± 38.33 μM) or human chymotrypsin (IC$_{50}$: 132.30 ± 10.57 μM) was more than 100 times weaker than that of 3CL$^{pro}$ from SARS-CoV-2 and SARS-CoV. These data indicate that myricetin has high selectivity toward 3CL$^{pro}$ from SARS-CoV-2 and SARS-CoV over other relevant proteinases.

**Mechanism of myricetin binding and reacting with the SARS-CoV-2 3CL$^{pro}$.** Inspired by the crystallographic observation that the side chain of His41 interacting with myricetin displays distinct orientation among most reported structures of the SARS-CoV-2 3CL$^{pro}$ (Supplementary Fig. 1b), we utilized Gaussian-accelerated molecular dynamics (GaMD) simulation to explore structural dynamics of His41 side-chain in the apo form of the protease. As shown in Supplementary Fig. 6a, the two identified configurations of His41 side chain could be represented by the angle of the imidazole ring projected on the backbone plane. Interestingly, during the simulation, the His41 side chain angle could be switched among the experimentally determined ones (Supplementary Fig. 6b, c) as the global structure of SARS-CoV-2 3CL$^{pro}$ maintains steady (Supplementary Fig. 7). Consequently, the His41 side chain does adopt two substantial configurations with the distribution slightly biased toward the one nearby Cys145 (with an angle of 97.6°) in the apo protease (Supplementary Fig. 6). It is thus reasonably speculated that for covalent inhibitors like myricetin, the structural transition of the intrinsically dynamic His41 side chain could afford space for the insertion of myricetin in the middle of His41 and Cys145, and meanwhile adopt a perfect side-chain conformation to stabilize the inhibitor by forming π–π stacking interactions with the chromone ring.

As it is the first time to observe that the pyrogallol group acted as a warhead to covalently bind with cysteine, quantum chemistry calculation was performed to understand the mechanism underlying the inhibitor-Cys145 covalent reaction and the effects of His41 (see the system models in Supplementary Fig. 8). It is well-known that polyphenols are susceptible to oxidation upon exposure to air[29]. To test whether myricetin reacting with GSH depends on oxygen, the rate of myricetin reacting with GSH in the absence of oxygen was measured. As shown in Fig. 3c, in the absence of oxygen only 11% myricetin was consumed after incubation with GSH for 600 min, while 65% myricetin was consumed in the presence of oxygen under the same condition. This result suggested that the auto-oxidation step is essential for the reactivity of myricetin with GSH or cysteine. Figure 4a schematically presents the reaction pathway for myricetin and GSH, including the auto-oxidation step of myricetin and the GSH adduction to the oxidated product of o-quinone. In the auto-oxidation step of a pyrogallol derivative in neutral or slightly alkaline solution, one hydroxyl group of the pyrogallol ring is first deprotonated to accelerate the electron oxidation to yield o-semiquinone radical[30], which decoys rapidly to form o-quinone[29]. The quantum calculation indicates no activation free energy barrier in the auto-oxidation (see the overall free energy change along the reaction path of GSH and myricetin in Supplementary Fig. 9). The subsequent adduction of GSH to o-quinone needs to overcome a high free energy barrier in neutral solution owing to the S–H bond cleavage of GSH (pKa = 8.66[31]; Fig. 4b). Such a high free energy barrier could be significantly decreased either by increasing pH of the solution such as introducing free OH$^{-}$ or by binding into the 3CL$^{pro}$ catalytic site and reacting with the His41-Cys145 dyad (Fig. 4b). In the latter case, myricetin o-quinone is stabilized by His41 in such a configuration that the sulfur atom of Cys145 S–H moiety can

attack o-quinone while the hydrogen of the Cys145 S-H group is allowed to be attracted by the deprotonated hydroxyl group of o-quinone (see the transition state (TS) in Fig. 4c), further supporting the covalent bond formation between the Cys145 S–H group and the myricetin o-quinone group. All the calculated TS geometries involved in the reactions were depicted in Supplementary Fig. 10.

**Structure-based design of myricetin derivatives and prodrugs.** The simple chemical structure, unique mode of action and good antiviral activity in vitro render myricetin valuable for further development. A structure-based chemical modification of myricetin was carried out. Considering the binding mode of myricetin with the protease and the synthetic ease, a methyl, ethyl, isoamyl, and cyclopentylmethyl group was introduced at the 7-OH of myricetin to obtain compounds **3**, **4**, **5**, and **6**, respectively (Fig 1a, Supplementary Fig. 11a and Supplementary Table 2). The addition of these alkyl groups to myricetin is also helpful to increase the cLogP of the compounds, since the cLogP of myricetin (0.84) is low (Supplementary Table 2). The introduction of the methyl group in compound **3** resulted in about twofold increase in the potency at the enzymatic level compared to myricetin (IC$_{50}$: 0.30 vs 0.63 μM, Fig. 1a, b and Supplementary Table 2) and about three-fold increase compared to baicalein (IC$_{50}$: 0.30 vs 0.94 μM). However, as the size of the substituted group increases, the inhibitory activity of the corresponding compounds decreases. The IC$_{50}$s of compounds **4**, **5**, and **6**, are 0.74, 1.92, and 2.45 μM, respectively (Supplementary Fig. 11a, b and Supplementary Table 2). The structure and activity relationship (SAR) of these derivatives suggests that the introduced alkyl group may bind to a specific but small sub-pocket which prefers the binding of a methyl group rather than other larger groups.

Subsequently, we determined the crystal structure of the SARS-CoV-2 3CL$^{pro}$ in complex with **3** (Fig. 2c and Supplementary Table 3), which was superimposed on myricetin as well as baicalein in the crystal structures of the protease for a comparison (Fig. 2 and Supplementary Fig. 12). As expected, the covalent bond is formed between Cys145 and the pyrogallol group of compound **3**, and the introduced methyl group binds into a small hydrophobic sub-pocket which is mainly constituted by residues Cys44/Met49/Pro52/Tyr54. In order to simultaneously maintain the covalent bonding and hydrophobic interactions, the chromone moiety of **3** has to rotate ~120 degree around the bond between the chromone and pyrogallol groups, resulting in a distinguishable orientation of **3** relative to that of myricetin (Fig. 2 and Supplementary Fig. 12a). As a result, the γ-pyrone rings in the chromone of compound **3** and baicalein are well-overlapped. Besides, a high overlap of the introduced methyl group of **3** with the free phenyl ring of baicalein is observed (Supplementary Fig. 12b). In other words, the sub-pocket holding the introduced methyl group of **3** is the S2 sub-site of 3CL$^{pro}$ which plays a key role in recognition of substrates as well as inhibitors like baicalein. In addition, multiple direct or water-mediated H-bonds were formed between the pyrogallol group of compound **3** and Thr26/Asn119/Gly143/Cys145, and between the chromone region of **3** and Glu166/Arg188/Gln189/Thr190 (Fig. 2c). Notably, the side-chain conformation of His41 in the **3**-bound complex differs from that in the myricetin-bound complex but is almost identical to that in the baicalein-bound complex. The π–π stacking interactions between the chromone region of myricetin or baicalein and His41 occurred for compound **3**. Accordingly, the binding pose of compound **3** in the protease is more similar to that of baicalein than its parent compound, myricetin (Fig. 2 and Supplementary Fig. 12).

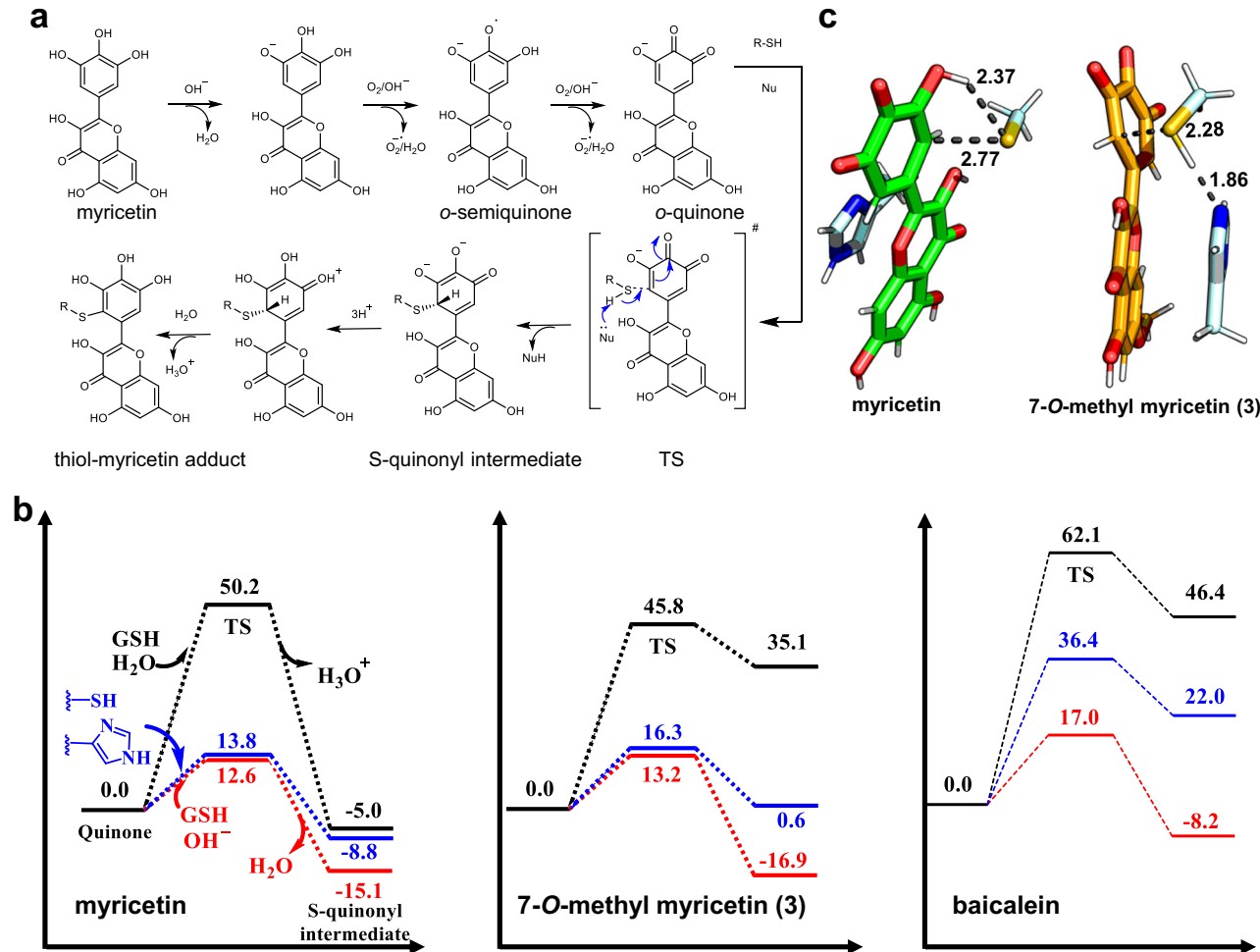

**Fig. 4 Mechanism of myricetin and its derivatives reacting with GSH or Cys145 of the SARS-CoV-2 3CL$^{pro}$. a** Proposed myricetin-GSH reaction pathway in aqueous solution (Nu: H$_2$O (neutral pH) or OH$^-$ (alkaline pH)). **b** Relative free energy profiles for the adduction of GSH or cysteine with $o$-quinone of myricetin, 7-$O$-methyl myricetin, and baicalein under different conditions (black: GSH in neutral pH solution, red: GSH in alkaline pH solution, blue: Cys145 in the SARS-CoV-2 3CL$^{pro}$). Values are given in kcal/mol. **c** The geometric difference between the transition states of myricetin and 7-$O$-methyl myricetin. Myricetin is shown in green sticks, compound **3** in orange sticks, and catalytic residues (His41 and Cys145) in palecyan sticks. Distances (angstrom) shown in dash lines suggest the existence of intermolecular interactions.

However, compared to baicalein, compound **3** inhibited the protease in a covalent manner, creating a covalent link to the catalytic Cys145, while baicalein used a non-covalent binding mode. The derivation of compound **3** from myricetin presents an example in which a minor chemical modification on the compound leads to a different binding pose.

The crystal structure of SARS-CoV-2 3CL$^{pro}$ in complex with **3** not only provides the molecular details of **3** recognition by the SARS-CoV-2 3CL$^{pro}$ but also explores the structural mechanism underlying the SAR of myricetin and its derivatives. Inspired by this, we also introduced a methyl group to the 7-OH of dihydromyricetin to generate 7-$O$-methyl dihydromyricetin (compound **7**) for the inhibitory activity test. The IC$_{50}$ of **7** is 0.26 μM, comparable to that of 7-$O$-methyl myricetin (compound **3**; Supplementary Fig. 11 and Supplementary Table 2). The crystal structure of SARS-CoV-2 3CL$^{pro}$ in complex with **7** was also determined (Fig. 2c and Supplementary Table 3), revealing a binding mode similar to that of **3**. These results together demonstrate that a congeneric series of myricetin is capable of inhibiting the proteolytic activity of 3CL$^{pro}$ via covalently targeting the catalytic cysteine of the protease.

The crystal structure also reveals that the side-chain conformation of His41 in the **3**- or **7**-bound complex is distinctive

from that in the myricetin-bound complex, leading us to perform the mechanism study as described above for the covalent binding of compound **3** with the two catalytic residues. Although 7-$O$-methyl myricetin has a free energy barrier at the similar level of myricetin in the covalent bond formation, it is facilitated by His41 in a different manner: the side chain of His41 is proximal to Cys145 and thus works as a nucleophile to attract the hydrogen of Cys145 S-H moiety (Fig. 4c). As a reference, the assumed reaction of Cys145 with the non-covalent inhibitor, baicalein, always requires much higher free energy barriers than myricetin or 7-$O$-methyl myricetin, implying the difficulty for baicalein to form covalent bond with Cys145 in 3CL$^{pro}$ (or GSH). Overall, these quantum chemistry calculation results are thus in well agreement with the experimental observations.

We further determined the antiviral efficacy of these compounds against SARS-CoV-2 in Vero E6 cells. The EC$_{50}$ values of **3** and **7** are 12.59 μM and 11.50 μM, respectively, similar to that of myricetin and dihydromyricetin. Although the IC$_{50}$ value of compound **3** against the SARS-CoV-2 3CL$^{pro}$ is about one-third of that of baicalein (IC$_{50}$: 0.30 vs 0.94 μM), its EC$_{50}$ value is larger than that of baicalein (EC$_{50}$: 12.59 μM vs 2.94 μM). The lower efficacy of **3** over baicalein is probably caused by the lower lipophilicity (cLoP: 1.48 vs 3.00), as compounds with a

higher lipophilicity are anticipated to have higher cell-membrane permeability. To our surprise, the $EC_{50}$ of 7-$O$-cyclopentyl-methyl-myricetin (compound **6**) also reaches a value of 7.56 μM, although its $IC_{50}$ against the SARS-CoV-2 3CL$^{pro}$ is weak in almost 10 times of **3** (Supplementary Table 2). Given a higher value of cLogP of **6** (3.57) over that of **3** (1.48), it is conjectured that the higher lipophilicity is more conducive for the compound to permeate the cell membrane. In other words, the high hydrophilicity of myricetin and compound **3** may result in a low cell-permeability and thus impede the antiviral activity of these compounds in the cell-based system.

Prodrug strategies are often used to improve the physicochemical, biopharmaceutical, or pharmacokinetic properties of pharmacologically potent compounds, while phosphate or phosphonate groups are the most common functional groups utilized to improve aqueous solubility or membrane permeability of compounds[32,33]. Accordingly, a proof-of-concept prodrugs (compounds **8** and **9**) were obtained by adding two kinds of phosphate groups (5,5-dimethyl-1,3,2-dioxayl phosphate and diphenyl phosphate) to the 7-OH of myricetin (Supplementary Fig. 1a and Supplementary Table 2) in order to improve the aqueous solubility as well as the membrane permeability of myricetin. The cLogP of two prodrugs is 2.26 and 3.89, respectively. In contrast to the weak inhibitory activities of these prodrugs against the SARS-CoV-2 3CL$^{pro}$ at the enzymatic assay, they do exhibit antiviral activities in the cell-based assays with an $EC_{50}$ of 33.45 and 3.15 μM, respectively (Fig. 1, Supplementary Fig. 11, and Supplementary Table 2). Compound **9** with the largest cLogP value displayed the most potent efficacy on the inhibition of the viral replication, demonstrating the reliability of the prodrug strategy and providing a good lead compound for further development. Inspired by the improved antiviral efficacy of compound **9**, the same phosphonate group was also added to the 7-OH of dihydromyricetin, resulting in compound **10**. The $EC_{50}$ of **10** against the replication of SARS-CoV-2 in the cells is 9.03 μM, better than that of dihydromyricetin (Fig. 1 and Supplementary Table 2). Therefore, the prodrug strategies afford a good opportunity for myricetin and its derivatives to improve the physicochemical or pharmacokinetic properties of the compounds.

**Compound 7 has the potential for oral administration.** Oral route of administration is the most convenient, common, and preferred for clinical therapy. Currently, most developed 3CL$^{pro}$ inhibitors are peptidomimetics and some of them displayed favorable pharmacokinetic (PK) profiling. However, these peptidyl inhibitors are hardly administrated orally, because amide bonds in these peptidomimetics are easily metabolized in vivo. Therefore, oral 3CL$^{pro}$ inhibitors are desired. We explored the oral PK profiling of myricetion and its derivatives. The results demonstrated that the PK profiling of compound **7** improved greatly compared to that of myricetin after an oral delivery at a dose of 30 mg/kg (Supplementary Table 4). When administered orally, compound **7** displayed an acceptable PK profile with a half time ($T_{1/2}$) of 1.74 h, an area under curve (AUC) of 510 ng h/mL, an acceptable oral bioavailability of 18.1%, a good maximal concentration ($C_{max}$) of 724 ng/mL, and a favorable plasma duration (MRT) of 1.89 h (Supplementary Table 4). A compound with oral bioavailability above 10% has a potential for development as an oral drug[15,34]. It is thus suggested that compound **7** has a prospect for oral administration. Further structural optimization to improve the PK profiling of the myricetin derivatives is ongoing with the aim of developing oral inhibitors of 3CL$^{pro}$.

## Discussion
Emerging CoVs like SARS-CoV, MERS-CoV, and SARS-CoV-2 cause globally prevalent and severe diseases in humans, raising great awareness about the increasing infection risks of highly pathogenic CoVs and calling for the development of efficacious anti-coronaviral drugs. 3CL$^{pro}$s are highly conserved cysteine proteases essential to the life cycle of CoVs, providing one of the most promising targets for antiviral agent development. The catalytic cysteine of 3CL$^{pro}$s presents one of the best nucleophiles for the design of covalently bound inhibitors. Accordingly, substrate analogs or mimetics attached with a chemical warhead targeting the catalytic cysteine were designed as peptidomimetic inhibitors of 3CL$^{pro}$s with a covalent mechanism of action[9]. Recently, we also reported the first crystal structure of the SARS-CoV-2 3CL$^{pro}$ in complex with a covalent peptidomimetic inhibitor (N3) identified by a mechanism-based strategy[13], and two peptidomimetic inhibitors which contain an aldehyde group acting as the warhead and exhibit excellent inhibitory activity as well as potent anti–SARS-CoV-2 infection activity[14].

Herein, we describe the state-of-the-art of the cysteine-directed chemical modification by the natural product and report a non-peptidomimetic covalent inhibitor of 3CL$^{pro}$s. The enzymatic assays, the crystal structure, the kinetic characterization, and the selectivity investigation clearly show that myricetin is a selective covalent inhibitor of the SARS-CoV-2 3CL$^{pro}$. The explored molecular mechanism suggests that the oxidized myricetin is first recognized by the catalytic site in which the specific side-chain conformation of His41 is prone to form the π–π stacking interactions with the chromone ring, which is followed by the covalent reaction of the pyrogallol moiety with Cys145. Moreover, the cell-based assay reveals that myricetin and its derivatives possess good inhibitory activity against the replication of SARS-CoV-2 in Vero E6 cells. In particular, the phosphate prodrug of myricetin (compound **9**) exhibits good antiviral efficacy. In addition, compound **7**, a derivative resulted from a small modification on myricetin, exhibited improved PK profiling compared to myricetin and highlighted the potential for oral administration.

Myricetin is a naturally occurring flavone observed in numerous edible plants, such as waxberries, oranges, grapes, herbs, and teas, and is one of the key ingredients of various foods and beverages[35,36]. A wide range of bioactivities of myricetin including the antioxidant, anticancer, antidiabetic, anti-inflammatory, and antiviral activities have been reported[36–38]. Here, the anti–SARS-CoV-2 effect of this natural product was revealed with strong evidence from the potent inhibitory activity data together with the crystal structure of SARS-CoV-2 3CL$^{pro}$ in complex with myricetin. With a huge natural resource, a simple chemical structure, low toxicity, and a unique mode of covalent action in targeting the SARS-CoV-2 3CL$^{pro}$, myricetin as well as its prodrug provides a preclinical candidate for further evaluation of its therapeutic potential in COVID-19.

Covalent ligands are of great interest as therapeutic drugs or biochemical tools. In the present study, the pyrogallol group of myricetin emerged as a warhead that could covalently linked to cysteine under the condition of oxidation. After the oxidation, the resulting $o$-quinone from the pyrogallol is also an α,β-unsaturated carbonyl group. Utilizing the pyrogallol group as a warhead and the chromone as the reversible binding portion, myricetin serves as a selective covalent inhibitor of cysteine proteinase. As many natural products contain the pyrogallol group, the explored reactivity of this group with the nucleophiles like cysteine also provides the vital clue for understanding the diverse bioactivities of natural products and identifying the phenolic natural products as covalent ligands. Moreover, the reactivity of the pyrogallol group can be modulated by the presence of oxygen or the change of pH. Accordingly, the pyrogallol group provides an alternative warhead with advantages for the development of covalent ligands or biochemical tools.

## Methods

**General chemistry**. Naturally occurring flavonoids for 3CL^pro inhibition test were from an in-house natural products library in Shanghai Institute of Materia Medica, Chinese Academy of Sciences. Myricetin (purity: 99.80%) and dihydromyricetin (purity: 99.14%) were purchased from Bide Pharmatech Ltd. Myricetin derivatives were synthesized and purified according to the general methods and procedures described in Supplementary Methods. The purities of the synthetic myricetin derivatives are over 95%. Analytical HPLC and ESIMS spectra were performed on a Waters 2695 instrument with a 2998 PDA detector coupled with a Waters Acquity ELSD and a Waters 3100 SQDMS detector using a Waters Sunfire RP C18 column (4.6 × 150 mm, 5 μm) with a flow rate of 1.0 mL/min. Masslynx was used to analyze the ESIMS data for all compounds. $^1H$ and $^{13}C$ NMR spectra were recorded on a Bruker AVANCE III 600 MHz instrument. Chemical shifts were reported in ppm (δ) coupling constants (J) in hertz. Chemical shifts are reported in ppm units with Me₄Si as a reference standard. NMR data for all compounds was performed on MestReNova.

**Protein expression and purification**. The cDNA of SARS-CoV-2 3CL^pro (Gen-Bank: MN908947.3) or SARS-CoV 3CL^pro (GenBank: AAP13442.1) was cloned into the pGEX6p-1 vector. To obtain the SARS-CoV-2 3CL^pro or SARS-CoV 3CL^pro with authentic N and C terminals, four amino acids (AVLQ) were inserted between the GST tag and the full-length SARS-CoV-2 3CL^pro or SARS-CoV 3CL^pro, while eight amino acid (GPHHHHHH) were added to the C-terminal of SARS-CoV-2 3CL^pro or SARS-CoV 3CL^pro. The plasmid was then transformed into BL21 (DE3) cells for protein expression. The N terminal GST tag and four amino acids (AVLQ) was self-cleavable. The expressed protein with authentic N terminal was purified by a Ni-NTA column (GE Healthcare) and transformed into the cleavage buffer (150 mM NaCl, 25 mM Tris, pH 7.5) containing human rhinovirus 3C protease for removing the additional residues. The resulting protein sample was further passed through a size-exclusion chromatography (HiLoad^TM 16/600 Superdex^TM 200 pg, GE Healthcare). The eluted protein samples were stored in a solution (10 mM Tris, pH 7.5) for the enzymatic inhibition assay, native state mass spectrometry studies, protein crystallization, etc.

The cDNA of full-length SARS-CoV-2 PL^pro (GenBank: MN908947.3) was cloned into the pET-22b vector. A cleavage site for the PreScission protease (LEVLFQGP) and 6His-tag were added to the C-terminus. The plasmid was then transformed into BL21 (DE3) cells for protein expression. The expressed protein was purified by a Ni-NTA column (GE Healthcare) and cleaved by the PreScission protease to remove the His-tag. The resulting protein sample was further passed through a size-exclusion chromatography (HiLoad^TM 16/600 Superdex^TM 200 pg, GE Healthcare). The eluted protein samples were stored in a solution (50 mM Tris pH 7.5, 100 mM NaCl, 10 mM DTT) for the enzymatic inhibition assay.

**Inhibition assays of SARS-CoV-2 3CL^pro, SARS-CoV 3CL^pro, SARS-CoV-2 PL^pro, and chymotrypsin**. A fluorescence resonance energy transfer (FRET) protease assay was applied to measure the inhibitory activity of compounds against the SARS-CoV-2 3CL^pro or SARS-CoV 3CL^pro. The fluorogenic substrate (MCA-AVLQSGFR-Lys(Dnp)-Lys-NH2) was synthesized by GenScript (Nanjing, China). The FRET-based protease assay was performed as follows. The recombinant SARS-CoV-2 3CL^pro (30 nM at a final concentration) or SARS-CoV 3CL^pro (100 nM at a final concentration) was mixed with serial dilutions of each compound in 80 μL assay buffer (50 mM Tris, pH 7.3, 1 mM EDTA) and incubated for 10 min. The reaction was initiated by adding 40 μL fluorogenic substrate with a final concentration of 20 μM. After that, the fluorescence signal at 320 nm (excitation)/405 nm (emission) was immediately measured every 30 s for 10 min with a Bio-Tek Synergy4 plate reader. The initial velocity of reactions added with compounds compared to the reaction added with DMSO were calculated and used to generate IC₅₀ curves.

The inhibition of SARS-CoV-2 PL^pro by compounds was measured with a fluorogenic peptide (RLRGG-AMC) synthesized by GenScript (Nanjing, China). The reactions were performed in a total volume of 120 μL. First, 50 nM SARS-CoV-2 PL^pro was incubated with the indicated concentrations of tested compounds under the condition of 50 mM HEPES, pH 7.5, 0.1 mg/mL BSA, and 5 mM DTT for 10 min. The reactions were initiated by the addition of 10 μM fluorogenic peptide. After that, the fluorescence signal at 360 nm (excitation)/460 nm (emission) was measured immediately every 1 min for 5 min with a Bio-Tek Synergy4 plate reader. The initial velocities of reactions with compounds added at various concentrations compared to the reaction added with DMSO were calculated and used to generate inhibition profiles.

The inhibition of chymotrypsin from bovine pancreas by compounds was carried out with a fluorogenic peptide (Suc-Leu-Leu-Val-Tyr-AMC) as substrate. The chymotrypsin (20 nM) was incubated with the indicated concentrations of tested compounds in 80 μL assay buffer (50 mM Tris, pH 7.3, 1 mM EDTA) and incubated for 10 min. The reactions were initiated by the addition of 40 μL substrate at a final concentration of 10 μM. After that, the fluorescence signal using 355 nm for excitation and 460 nm for emission was immediately measured every 50 s for 5 min with a Bio-Tek Synergy4 plate reader. The initial velocity of reactions added with compounds compared to the reaction added with DMSO were calculated and used to generate IC₅₀ curves.

For each compound, three independent experiments and each independent experiment in duplicate were performed for the determination of IC₅₀ values. At least nine concentrations of a compound were used to calculate IC₅₀ values. The final concentration of DMSO is <2% of the total volume, which had no effect on the enzyme activity of SARS-CoV-2 3CL^pro, SARS-CoV 3CL^pro, SARS-CoV-2 PL^pro, and chymotrypsin. The IC₅₀ values were expressed as the mean ± SD and determined via nonlinear regression analysis using GraphPad Prism software 8.0 (GraphPad Software, Inc., San Diego, CA, USA).

**Protein crystallization and structure determination**. The purified SARS-CoV-2 3CL^pro protein was concentrated to 7 mg/mL for crystallization. The apo SARS-CoV-2 3CL^pro crystals were grown at 20 °C by mixing equal volumes of protein and a reservoir (12% PEG6000, 100 mM MES, pH 6.0, 3% DMSO) with a handing-drop vapor diffusion method. To obtain complex structures, the SARS-CoV-2 3CL^pro protein was incubated with 5 mM myricetin (**1**), compound **3**, or compound **7** for 1 h before crystallization condition screening. Crystals of the complexes were obtained under the condition of 10–22% PEG6000, 100 mM MES, pH 5.75–6.25, and 3% DMSO. Crystals were flash frozen in liquid nitrogen in the presence of the reservoir solution supplemented with 20% glycerol. X-ray diffraction data were collected at beamline BL18U1 at the Shanghai Synchrotron Radiation Facility[39]. Bluice was used to collect X-ray diffraction data. The data were processed with HKL3000 software packages[40]. The complex structures were solved by molecular replacement using the program PHASER[41] with a search model of PDB code 6LU7. The model was built using Coot[42] and refined with XYZ (reciprocal-space), Individual B factors, TLS parameters, and Occupancies implemented in the program PHENIX[43]. The refined structures were deposited to Protein Data Bank with accession codes listed in Supplementary Table 3. The complete statistics as well as the quality of the solved structures are also shown in Supplementary Table 3. All structural figures were generated using Pymol.

**Cell-based antiviral activity assay**. The Vero E6 cell line was obtained from American Type Culture Collection (ATCC, Manassas, USA) and maintained in minimum Eagle's medium (MEM; Gibco Invitrogen) supplemented with 10% fetal bovine serum (FBS; Invitrogen, UK) in a humid incubator with 5% CO₂ at 37 °C. The cytotoxicity of tested compounds on the Vero E6 cells were determined by CCK8 assays (Beyotime, China). A clinical isolate SARS-CoV-2[2] was propagated in the Vero E6 cells, and the viral titer was determined by 50% tissue culture infective dose (TCID₅₀) using immunofluorescence assay[44]. All the infection experiments were performed at biosafety level-3 (BSL-3).

Pre-seeded Vero E6 cells (2 × 10⁵ cells/mL) were incubated with different concentrations of the compounds for 1 h and the virus was subsequently added (a multiplicity of infection of 0.01) to infect the cells for 2 h. The final concentration of DMSO is <0.1% of the total volume. After that, the virus-compound mixture was removed and cells were further cultured with a fresh compound containing medium. At 24 h post infection, the cell supernatant was collected and the viral RNA in supernatant was subjected to qRT-PCR analysis of the copy numbers of the receptor binding domain (RBD) of SARS-CoV-2 spike protein[44]. The primers used for qRT-PCR were RBD-qF1: 5′-CAATGGTTTAACAGGCACAGG-3′ and RBD-qR1: 5′-CTCAAGTGTCTGTGGATCACG-3′ (Supplementary Table 5). Six independent experiments (each experiment in triplicate) were performed for myricetin, dihydromyricetin, **3**, **7**, **9**, and **10**, and three independent experiments (each experiment in triplicate) were performed for **4**, **5**, **6**, and **8**. Six concentrations of each compound were used to calculate EC₅₀ values. The EC₅₀ values were expressed as the mean ± SD.

**Kinetic analysis**. The interaction of myricetin with the SARS-CoV-2 3CL^pro can be described in two steps according to Eq. (1), an initial reversible binding event followed by formation of the covalent bond:

$$3CL^{pro} + I \overset{K_i}{\rightleftharpoons} 3CL^{pro} \cdot I \xrightarrow{k_{inact}} 3CL^{pro} - I \qquad (1)$$

The reversible binding equilibrium is determined by $K_i$, the first-order rate constant of the reaction step is $k_{inact}$. For determination of $K_i$ and $k_{inact}$, 100 nM recombinant SARS-CoV-2 3CL^pro was incubated with 2.5–40 μM myricetin for 13–243 s. At each time point, the FRET protease assay was applied as mentioned above. Relative protease activity for various inhibitor concentrations over a time course were fit to an exponential equation to generate $k_{obs}$ values for each concentration tested. With three independent experiments, the resulting $k_{obs}$ values were then plotted versus inhibitor concentration, and $k_{inact}$ and $K_i$ values were generated according to the equation:

$$k_{obs} = k_{inact} \left( \frac{[I]}{[I] + K_i} \right) \qquad (2)$$

The overall potency is described by the second-order rate constant $k_{inact}/K_i$.

**Half-life determination of myricetin reacting with GSH**. Half-life determination of myricetin reacting with GSH was conducted according to the method developed by Flanagan et al.[27]. Briefly, 400 μM of myricetin was incubated with 10 mM GSH for 120, 240, 360, 480, and 600 min (in the presence or absence of oxygen), respectively. As a positive control, N-phenylacylamide was incubated with 10 mM

GSH for 20, 30, 40, 60, and 90 min, respectively. The hydrochloric acid was added at a final concentration of 100 mM to terminate the reaction. The remaining myricetin or $N$-phenylacylamide at different conditions was determined by LC/MS with indoprofen as internal standard in mass spectrometry analysis. The data was analyzed with Analyst software. Ln (the percentage of the remaining myricetin or $N$-phenylacylamide) was plotted against incubation time to generate the half-life time of myricetin or $N$-phenylacylamide reacting with GSH.

**Gel-based competitive ABPP assay.** HEK293T cells were maintained with Dulbecco's Modified Eagle Medium (Gibico) supplemented with FBS (Invitrogen, UK) in a humid incubator with 5% $CO_2$ at 37 °C. The cells were digested with 0.05% trypsin (Invitrogen, UK) and washed twice with phosphate-buffered solution (PBS). Afterwards, the cell pellet was resuspended with a cold lysis buffer containing 50 mM Tris (pH 7.5), 150 mM NaCl, and 2% Triton followed by an incubation in ice for 20 min. After a centrifugation at $17,226 \times g$ for 20 min, the supernatant was collected and stored at −80 °C. Protein concentrations were determined with the Bradford protein assay. The recombinant SARS-CoV-2 3CL^pro (l μg/mL), the HEK293T lysate (0.2 mg/mL), or the mixture of these two was pre-incubated with the vehicle or different concentrations (0.5, 2, and 10 μM) of myricetin at room temperature for 20 min followed by an incubation with Alexa Fluor™ 488 C5-maleimide (No.2096405, Invitrogen) at a final concentration of 1 μM for 20 min at room temperature. The final concentration of DMSO is <0.5% of the total volume. Samples were resolved on 12.5% acrylamide SDS-PAGE gel and visualized by in-gel fluorescence scanning (Typhoon FLA 9500, GE Healthcare).

**Molecular dynamics simulation.** It has been revealed that the monomeric form of 3CL^pro is catalytically inactive and the dimer structure is the prerequisite for the enzymatic activity performance of the protease[12,45], the structural dynamics of the SARS-CoV-2 3CL^pro dimer instead of monomer was investigated here by using GaMD. GaMD is a sophisticated enhanced sampling MD simulation method which has been extensively applied in a variety of biomolecular simulations for protein folding, protein conformational transition, and protein-ligand binding[46–52]. Detailed information of GaMD has been previously described in the literature[46,47].

The atomic coordinates of apo 3CL^pro dimer were retrieved from the Protein Data Bank (PDB code: 6M2Q) with the crystal water molecules maintained. The protonation states of all titratable residues at pH 7.4 were determined using H++ web service[53], consistent with the standard AMBER protonation states at physiological pH. Particularly, all His residues stayed at the neutral (deprotonated) states but displayed different hydrogen additions. For example, proton presented at the HD1 position of His164 but at the HE2 position in His41, His163, and His172 at the active site. After that, the 3CL^pro dimer was solvated in a $98 \times 105 \times 90$ Å³ cubic box filled with a total of 22,314 water molecules. Multiple Na$^+$ ions were added to neutralize the protein charges. AMBER 18 suite of program[54] was employed for simulation with the underlying force fields of FF99SBildn force field[55] for protein and TIP3P model[56] for water.

The constructed system was initially minimized for 50,000 steps and heated to 300 K, with the protein heavy atoms being fixed using a harmonic restraint with the force constant of 10.0 kcal mol$^{-1}$ Å$^{-2}$. Subsequently, the protein was relaxed by two steps of equilibrium at constant temperature of 300 K and constant pressure of 1 atm (*NPT* ensemble): 2 ns for relaxing protein side chain and 2 ns for protein main chain. The shake algorithm implemented in Amber 18 was used to fix all covalent bonds involving hydrogen atoms and periodic boundary conditions were used to avoid edge effects[57]. The Particle Mesh Ewald method was applied to treat long-range electrostatic interactions and the cutoff distance for long-range terms (electrostatic and van der Waals energies) was set as 8.0 Å[58]. The Langevin dynamics with a collision frequency of 2.0 ps$^{-1}$ was adopted to control the temperature. Finally, the GaMD simulations were performed on the equilibrated system using the GaMD module implemented in the GPU version of AMBER 18, including a 12-ns short conventional MD simulation for collecting the potential statistics to define GaMD acceleration parameter values, a 12-ns equilibration after adding the boost potential, and finally two independent ~1.5 μs GaMD production simulations with randomized initial atomic velocities. All GaMD simulations were run at the "dual-boost" level by setting the reference energy to the lower bound, one boost potential being applied to the total potential and the other to the dihedral energetic term. The average and the standard deviation (SD) of the system potential energies were calculated every 300,000 steps (0.6 ns). The upper limit of the boost potential SD was set to 6.0 kcal/mol for both the dihedral and the total potential energetic terms. The coordinates were saved every 10,000 steps.

**Ab initio calculation.** The ab initio calculation was carried out using the Gaussian 09 program[59]. In all, 9 systems (myricetin, 7-$O$-methyl myricetin and baicalein (control test) reacting with GSH (in neutral or alkaline solution) or with His41 and Cys145) were prepared. To mimic the reaction pathway in the protein environment, each relevant system was truncated as a model shown in Supplementary Fig. 8 and the boundary atoms were fixed at their positions inside the protein, ensuring that each reaction moiety stayed in the similar orientation as that in the protein environment. Additionally, to capture the transition states, the crystal

geometries of inhibitors were slightly adjusted fulfilling the Burgi–Dunitz criteria of near-attach-conformation parameters: the distance between the sulfur atom of Cys145 and the carbon atom of inhibitors in the attacking state (S---C) <3.5 Å and the attacking angle of ($105 \pm 5$) degree. For each abovementioned system, geometry optimization was conducted at the M06-2X/6-311++G(d, p) level to generate the optimized (lowest energy) geometry, frequency analysis at the same level was performed to confirm the obtained geometry as a local energy minimum or a transition state, and to achieve the thermal correction to the Gibbs free energy[60,61]. Single-point energy calculation was carried out on the optimized geometry with the same basis set, 6-311++G(d, p). The SMD (Solvent Model based on Density) method for water (default) was used to incorporate solvent effects[62]. Finally, the Gibbs free energy was obtained by adding the thermal correction to the single-point energy. The calculated energies of all involved substances are summarized in Supplementary Tables 6−9.

**PK Study of myricetin and 7 in mice.** Six-week-old ICR male mice were housed in a 12/12-h light/dark cycles at 25 °C and humidity 40–70% with regular chow diet and free access to water. At least six mice, weighting 18−22 g each were randomly divided into two groups. Compound **7** (or myricetin) dissolved in water containing 5% DMSO and 0.5% hydroxypropyl methyl cellulose (HPMC) was administered orally at a dose of 30 mg/kg. Blood samples at seven time points (0.15, 0.3, 1.0, 2.0, 4.0, 8.0, and 24 h) were collected. Another group of at least three mice were given intravenously of compound **7** with a single dose (10 mg/kg) dissolved in ethanol/ PEG300/saline (10/40/50, v/v/v/v). Blood samples at seven time points (0.03, 0.15, 0.75, 2.0, 4.0, 8.0, and 24 h) were also collected. Plasma concentrations of **7** were analyzed using an AQUITY UPLC system with a thermostatted autosampler and an ultrahigh performance binary pump (I-class, Waters, MA, USA), and a triple quadrupole mass spectrometer with electrospray ionization (ESI) source (Xevo TQ-S, Waters, MA, USA).

All animal experiments were performed following animal ethics guidelines and protocols approved by the Institutional Animal Care and Use Committee of Shanghai Institute of Materia Medica (Accreditation number: 2020-02-YY-11).

**Reporting summary.** Further information on research design is available in the Nature Research Reporting Summary linked to this article.

## Data availability
The atomic coordinates and structure factors have been deposited into the Protein Data Bank with accession codes 7DPP (SARS-CoV-2 3CL^pro in complex with myricetin), 7DPU (SARS-CoV-2 3CL^pro in complex with **3**), and 7DPV (SARS-CoV-2 3CL^pro in complex with **7**).

All data are available from the corresponding author upon reasonable request.

The cDNA of SARS-CoV-2 3CL^pro and PL^pro (GenBank: MN908947.3) or SARS-CoV 3CL^pro (GenBank: AAP13442.1) were obtained from Genbank (https://https.ncbi.nlm. nih.gov/genbank/). Source data are provided with this paper.

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

## Acknowledgements

We thank Prof. H. Eric Xu and Prof. Huixiong Dai at Shanghai Institute of Materia Medica, Chinese Academy of Sciences, for their constructive comments and the staff from beamlines BL17U1 and BL18U1 at Shanghai Synchrotron Radiation Facility. This work was supported by the National Key R&D Program of China (No. 2016YFA0502301 and 2017YFB0202604), the National Natural Science Foundation of China (No. 21877122, No. 91953000, No. 32071248, and No. 21920102003), the Science and Technology Commission of Shanghai Municipality (No. 20430780300 and No.

Nature Communications
The Author(s) 2021

18430712500), and the International Partnership Program of Chinese Academy of Sciences (153631KYSB20160004, 153631KYSB20170043). Computational resources were provided by Tianhe II supercomputer in Guangzhou, China.

## Author contributions

H.S., H.X., and W.Z. parepared the protein sample and performed the enzymatic assay and Gel-Based Competitive ABPP Assay. H.S., W.Z., and M.L. dertermined the crystal structure. S.Y. and C.K. parepared the compounds and determined the structures of myricetin and dihydromyricetin derivatives. J.L. conducted MS/MS analysis of myricetin covantly binding to 3CL$^{pro}$ and in vivo PK Study of compounds in mice. Q.S. explored molecular mechanism of myricetin and its derivatives reacting with GSH or Cys145 of SARS-CoV-2 3CL$^{pro}$. H.S., W.Z., and F.L. carried out the quantitative analysis in the determination of GSH$_{t1/2}$ of myricetin. Y.Z., Q.W., L.Z., W.S., and G.X. performed antiviral activities measurment in cells. X.J. and J.S. helped with data analysis and interpretation. H.J., Y.X., Y.Y. and L.Z. initiated the project and supervised the research. Y.X. wrote the manuscript with input from all co-authors.

## Competing interests

The authors declare no competing interests.
