## [Peer Review File · Nature Communications]

REVIEWER COMMENTS

Reviewer #1 (Remarks to the Author):

This paper discovered one flavonoid as the covalent inhibitor of SARS-CoV-2 3CLpro, which could function as antivirals against SARS-CoV-2 with a novel mechanism. The mechanism of myricetin and its derivatives reacting with GSH or Cys145 is also elucidated in details. And it also did a structure-based design and optimization of the potential prodrugs. It also provides the platform to design potential and promising drugs against COVID. I have some comments as the following.

- (1) The target of myricetin or remdesivir is different. It is not easy to compare their efficacy without animal assays.
- (2) The paper said that Glu189 forms two hydrogen bonds with the chromone moiety of myricetin. Please check figure 2a. It should be clearly described.
- (3) What is the definition of non-bonding interactions? (In line 172). Is hydrogen bonding a non-bonding interaction?
- (4) For the figure 2a and b, there is no need to show the electron density map for the compounds alone. It could be put into the cartoon and stick-bond in a or b.
- (5) Since the binding mode of 3 is more similar to baicalein. Their EC50 or XXX should be also compared. And 3 could form covalent bond with Cys145, but baicalein could not. This is the essential difference.
- (6) In line 565, it should be pGEX6p-1.
- (7) In line 640, Phenix program refines the structure not only with simulated-annealing protocol. At this resolution, you may have used individual B-factor refinement and also maybe TLS.
- (8) In line 661, are two independent experiments good enough? Normally, there should be at least six independent experiments to get solid data to provide the conclusion.
- (9) For the MD part, there should be explanation for the use of 3CL dimer instead of monomer in the GaMD.
- (10) In the crystallography data collection and refinement table, for the 3CLpro-3 and 7, the spacegroup is P12(1)1 or P121? It should be clearly written.
- (11) The compound Apigenin showed -1.0% inhibition and Luteolin showed -4.0% inhibition, and Isorhamnetin -2.6%. Maybe they could activate 3CLpro instead of inhibition?

Reviewer #2 (Remarks to the Author):

In this article, Su et al. reported a flavonoid natural product, i.e. myricetin, as a SARS-CoV-2 3CLpro/Mpro covalent inhibitor with pyrogallol as a novel electrophile warhead attacking the thiol group of the catalytic Cys 145. The authors first screened a panel of flavonoid compounds against Mpro and identified two hits, including myricetin and Dihydromyricetin. Following the measurement of low-micromolar range IC50 and antiviral EC50 values for Myricetin and its derivatives, co-crystal structures were further determined for myricetin, comp.3 and 7 at high resolutions (better than 2.5Å). Based on the electron density maps and mass spec results, the authors discovered the covalent mechanism of action for these myricetin derivatives. Surprisingly, Cys145 of Mpro covalently conjugated to the 2'-carbon of myricetin. Therefore, the authors employed quantum chemistry calculation to understand the mechanism of this reaction. In addition, the authors also worked on phosphonate derivatives of myricetin as prodrugs in order to improve their PK and PD properties.

Since the outbreak of COVID-19, quite a few covalent inhibitors have been reported for SARS-CoV-2 Mpro so far. The authors previously published several Mpro inhibitor papers (Jin et al., Nature, 2020 and Dai et al., Science, 2020). In comparison with their own papers in Nature and Science, these myricetin derivatives are all at the same pocket of Mpro and share a covalent nature. Therefore, these myricetin derivatives are not new in terms of mechanisms. In comparison with another flavonoid non-covalent inhibitor Baicalein, reported by the same authors (ref 24), myricetin takes the same binding

site but with a covalent mechanism. Because some covalent inhibitors have shown superior potency, e.g. GC376 ($k_i=60$ nM, Ma et al., Cell Research, 2020), myricetin and its derivatives are not best-in-class inhibitors either.

However, as the authors mentioned, this article might be the first reporting pyrogallol as a novel electrophile warhead attacking the catalytic Cys of any Cysteine proteases. The mechanism for the conjugation of Cys145 SH to the 2' carbon of pyrogallol is novel. Given the severity of the global COVID-19, this novel natural product covalent inhibitor for SARS-CoV-2 Mpro with an unexpected mechanism can potentially be of interest to the readers of Nature Communications.

Major points:

1. To better understand the intrinsic reactivity and gain insights into the mechanism, can the authors provide in vitro reactivity assessments of inhibitors with some sulfur nucleophiles such as GSH or N-Acetyl-L-cysteine (simple model compound to mimic cysteines in proteins) as suggested in literatures (J. Med. Chem. 2014, 57, 10072–10079; Med. Chem. Commun., 2016, 7, 864–872)? Can the authors use MS or NMR or another approach to demonstrate the existence of the covalent adducts or trapped reaction intermediates in the reaction model proposed in Fig. 4c?
2. Flavonoids are promiscuous. The authors investigated the selectivity of myricetin over cysteine proteases. How about over other proteins or enzymes? There are numerous PDB structures for flavonoids and/or myricetin bound kinases, e.g. PDB 6M88 and 2O63.
3. How important is HIS41 in the covalent bond formation between myricetin and C145? Can the authors show experimentally what will happen if HIS41 is mutated, in terms of myricetin inhibitory activity and covalent adduct formation with Mpro.
4. Line 661, “at least two independent experiments were carried out for each compound”, would duplicates for EC50 measurement be acceptable for nature communications?

Minor points:

1. Line 48, “three global pandemics”, SARS and COVID-19 have been declared by WHO as pandemic. Is MERS too?
 2. Line 81, ref 12 is a research paper about neutralizing antibody for the spike protein of SARS-CoV-2, did it show that Mpro is one of the best targets for CoVs? I would suggest the authors double check.
 3. Ref 13 and ref 31 are the same paper?
 4. Line 139, PCR was used for EC50 measurement, PCR for the expression of what protein? What primers have been used?
 5. Line 358, IC50 increase from 0.31 to 0.67 μ M, is this a one-fold increase or two?
 6. Line 398, IC50s of comp 7 and 3 are 0.26 and 0.31 μ M respectively (based on Suppl Table 2), these two are essentially the same, no one is better than the other one. Why it was stated “even better than”?
 7. Line 691, -80 XXX?
 8. Line 708, PDB 6M2N was used in the calculation, 6M2N is the baicalein bound Mpro. why not the myricetin-bound structure used in the calculation, e.g. 7DPP or 7DPU?
 9. In supplementary materials, suppl figures 9 and 10 were missing in table of contents, line 24
 10. Some grammars, I just list a few but not all. All authors should proof-read through the article.
- Line 271, was added to labeled  was added to label
Line 328, owe to  owing to
Line 387, are not occurred  did not occur
Line 515, new warhead that could covalent modification of

Reviewer #3 (Remarks to the Author):

The manuscript "Identification of pyrogallol as a novel warhead in the design of covalent inhibitors for the SARS-CoV-2 3CL protease" reports the identification of myricetin and derivatives as covalent inhibitors of SARS-CoV-2 3CL protease. This study includes experimental characterization of compound potency, selectivity, determination of binding mode by crystallography, and antiviral activity. Computational studies (molecular dynamics simulations and quantum chemistry calculations) have also been performed to study the covalent reaction mechanism and energies involved. There is a clear contribution of the study in terms of characterizing a new class of SARS-CoV-2 3CL protease inhibitors, in a fairly complete study. Thus, this study could potentially be appropriate for publication in Nature Communications. However, I recommend improvement of the following points before a new evaluation of this manuscript.

- Figure 4 – The schematic panel in Figure 4c is not very informative. Since the conformation of the catalytic His is discussed as important to define whether Cys145 reacts with each compound, it would be better to have tridimensional figures clearly illustrating the different conformations of His41 in these simulations.

- Lines 388-392 – Based on Supplementary Figure 9, in my opinion, this is not a convincing assertion. There is indeed a rotation in the ring, but it does not constitute "a surprisingly different" binding mode. I also disagree with emphasizing that potency is improved, based on only a two-fold difference in IC50 values. This is a minor potency improvement.

- Lines 461-467. Please cite references that support the discussion that the PK profile for compound 7 is acceptable.

- Lines 474-475. Please clarify why Cys is considered an "intriguing amino acid for the design of targeted covalent inhibitors". To my knowledge, Cys is recognized as one of the best nucleophiles and it is likely the most exploited residue for the design of covalently bound inhibitors.

- The current conclusion is very long and contains much text which does not refer to the conclusions of this manuscript. Some content is also repetitive, as in the description of pyrogallol as a novel warhead.

- Lines 483-485. Dozens of structures of SARS-CoV-2 Mpro bound to covalent inhibitors are available. Thus, it cannot be said that this manuscript is the first report of small molecule covalent inhibitors of this enzyme.

- Lines 494-497 – Can the antiviral activity of compound 9 be considered "excellent"? And is the PK profile of compound 7 "greatly improved"? Based on the current manuscript and data provided, these assertions from the authors are not clear to me.

- Lines 500-503. This is not the first report of antiviral activity for myricetin and derivatives. Please see publications such as: DOI: 10.1080/14756366.2020.1754813, DOI: 10.1016/j.antiviral.2020.104714.

- In the methods session, please add more information on the following topics:

- What was the minimum purity of the compounds synthesized?
- Please provide the gene accession code that allows the reader to obtain the exact sequence of the Mpro and PLpro expressed.
- Please indicate the maximum amount of DMSO used in the assays and inform what is the impact of such DMSO concentration in the activity of the enzymes. Please report the minimum number of compound concentrations for each IC50 curve and the number of replicates in each independent experiment.
- In the molecular dynamics protocol, please mention which protonation states were predicted for the titratable active site residues, when using H++.

- Supplementary Tables 4-7 provide details on calculated energies from quantum mechanical calculations. However, these tables are not discussed anywhere in the manuscript. It is important to discuss these results, at least in the supplementary material.

Some additional minor points for improvement:

- Lines 432-434 - The difference between the original hits and compounds 8 and 9 is not just phosphorylation. Please clarify the text to give more precise information about the substituent added to these derivatives.
- In Figures 2, the color scheme with carbons in magenta makes it very difficult to distinguish Carbons from Oxygens (red). Please employ a color scheme with better contrast between Carbon and Oxygen.
- In Supplementary Figure S1, the PDB codes listed do not correspond to all available SARS-CoV-2 Mpro structures. Please modify the figure legend or add other structures to the figure. Also, highlight/clarify which structure is bound to baicalein, since the manuscript references this figure when discussing binding to this compound.

Sincerely,

Prof. Dr. Rafaela Salgado Ferreira

Associate Professor

Department of Biochemistry and Immunology

Federal University of Minas Gerais (UFMG), Brazil

We want to thank the referees for their positive comments, constructive suggestions and thorough review of this work. We have revised our manuscript to fully address all comments and suggestions made by three reviewers. Below are our point-by-point responses (colored **blue**) to the Reviewers' comments (colored **black**). For your convenience, all the changes in the text made in response to the comments have been highlighted in **red** in the revised manuscript.

Responses to Reviewer 1:

Comments: This paper discovered one flavonoid as the covalent inhibitor of SARS-CoV-2 3CLpro, which could function as antivirals against SARS-CoV-2 with a novel mechanism. The mechanism of myricetin and its derivatives reacting with GSH or Cys145 is also elucidated in details. And it also did a structure-based design and optimization of the potential prodrugs. It also provides the platform to design potential and promising drugs against COVID. I have some comments as the following.

Response: We are grateful for the positive comments by the reviewer.

Comments: The target of myricetin or remdesivir is different. It is not easy to compare their efficacy without animal assays.

Response: We agreed with the reviewer that the target of myricetin or remdesivir is different. Remdesivir, a potent inhibitor of the SARS-CoV-2 RdRp, is the first small molecule drug approved by FDA for the treatment of COVID-19. so, we used it as a positive control in our study and made it clear in the revised manuscript (Lines 142-143, Page 8). In addition, following the reviewer's suggestion, we have removed the misleading description relevant to the efficacy comparison between remdesivir and the derivatives of myricetin from the manuscript (Line 39, Page 3; Line 114, Page 7; Lines 143-144, Page 8; Lines 450-453, Page 25).

Comments: The paper said that Glu189 forms two hydrogen bonds with the chromone moiety of myricetin. Please check figure 2a. It should be clearly described.

Response: In the revised manuscript, it has been changed into “The chromone moiety of myricetin established H-bonds with the side chain of Glu189 as well as a buried water molecule which simultaneously contacted with His164/His41/Asp187.” (Lines 167-172, Page 10).

Comments: What is the definition of non-bonding interactions? (In line 172). Is hydrogen bonding a non-bonding interaction?

Response: Non-bonding (non-bonded) interactions act between atoms which are not linked by covalent bonds. It includes hydrogen bonding Interactions, hydrophobic interactions, ionic Interactions, and so on (Prog. Chem. Org. Nat. Prod. 2019, 11: 99-141, DOI: 10.1007/978-3-030-14632-0_4). To avoid confusion, we have changed the non-bonding interactions into “noncovalent interactions” (Line 174, Page 10). The revised sentence is “myricetin is perfectly engaged with the catalytic site by making both covalent bonding and noncovalent interactions with the surrounding residues.”

Comments: For the figure 2a and b, there is no need to show the electron density map for the compounds alone. It could be put into the cartoon and stick-bond in a or b.

Response: Following the reviewer’s suggestion, the electron density maps for baicalein and myricetin have been put into the cartoon and stick-bond in figure 2a and 2b, respectively, in the revised manuscript (Page 12).

Comments: Since the binding mode of 3 is more similar to baicalein. Their EC50 or XXX should be also compared. And 3 could form covalent bond with Cys145, but baicalein could not. This is the essential difference.

Response: Thank the reviewer for the constructive suggestion. We have added the content comparing the IC₅₀S, EC₅₀S, and binding modes between baicalein and **3** in the revised manuscript (Lines 362-363, Page 21; Lines 390-394, Page 22 and Lines 422-427, Page 24). The binding pose of compound **3** in the protease is more similar to that of baicalein than its parent compound, myricetin. However, compared to baicalein, compound **3** inhibited the protease in a covalent manner, creating a covalent link to the catalytic Cys145, while baicalein used a noncovalent binding mode. Although the IC₅₀ value of compound **3** against the SARS-CoV-2 3CLpro is one-third of that of baicalein (IC₅₀: 0.30 vs 0.94 μM), its EC₅₀ value of **3** is larger than that of baicalein (EC₅₀: 12.59 μM vs 2.94 μM). The lower efficacy of **3** over baicalein is probably caused by the lower lipophilicity (cLoP: 1.48 vs 3.00), as compounds with a higher lipophilicity are anticipated to have higher cell-membrane permeability.

Comments: In line 565, it should be pGEX6p-1.

Response: It has been corrected in the revised manuscript (Line 572, Page 31).

Comments: In line 640, Phenix program refines the structure not only with simulated-annealing protocol. At this resolution, you may have used individual B-factor refinement and also maybe TLS.

Response: We refined with XYZ (reciprocal-space), Individual B factors, TLS parameters, and Occupancies implemented in the program PHENIX. It has been corrected in the revised manuscript (Lines 651-652, Page 34).

Comments: In line 661, is two independent experiments good enough? Normally, there should be at least six independent experiments to get solid data to provide the conclusion.

Response: Following the reviewer's suggestion, we carried out more independent measurement on each compound (Lines 675-678, Page 36). In

the revised manuscript, a total of six independent experiments (each experiment in triplicate) were performed for myricetin, dihydromyricetin, **3**, **7**, **9**, and **10** which are presented in the main text. Three independent experiments (each experiment in triplicate) were performed for compounds **4**, **5**, **6**, and **8** shown in supplementary information.

Comments: For the MD part, there should be explanation for the use of 3CL dimer instead of monomer in the GaMD.

Response: Thank the reviewer for the suggestion. The reason for the use of 3CL dimer instead of monomer has been added in the revised manuscript (Lines 722-725, Page 38). The monomeric form of 3CLpro is catalytically inactive while the dimeric structure is the prerequisite for the enzymatic activity performance of 3CLpro. Therefore, the structural dynamics of the SARS-CoV-2-3CLpro dimer instead of monomer was investigated here by using Gaussian accelerated MD (GaMD).

Comments: In the crystallography data collection and refinement table, for the 3CLpro-3 and 7, the space group is P12(1)1 or P121? It should be clearly written.

Response: The space group is P1 2(1) 1. It has been corrected in the revised Supplementary Information (Supplementary Table 3, Page 26 in Supplementary Information).

Comments: The compound Apigenin showed -1.0% inhibition and Luteolin showed -4.0% inhibition, and Isorhamnetin -2.6%. Maybe they could activate 3CLpro instead of inhibition?

Response: Inhibitory activities of the compounds against the SARS-CoV-2 3CLpro were determined using the fluorescence resonance energy transfer (FRET) protease assay. The inhibition ratio of each compound at various concentrations were calculated by comparing the velocity of each reaction (V_{test})

with that of the reaction added with DMSO ($V_{control}$) according to the formula: Inhibition Ratio = $1 - (V_{test}/V_{control})$. If compound A have no influence on the activity of the SARS-CoV-2 3CLpro, the V_{test} is equal to $V_{control}$ and the inhibition ratio is equal to zero theoretically . However, in practice, the measured V_{test} may be a little bit larger or smaller than the measured $V_{control}$ due to the reasons like a random error resulting from pipette volume error or reading error of microplate reader. The measured inhibition ratios of Apigenin, Luteolin and Isorhamnetin was -1.0%, -4.0% and -2.6%, respectively, which is close to zero. It thus means that these three compounds nearly have no influence on the enzyme activity of the SARS-CoV-2 3CLpro at the concentration of 10 μ M.

Responses to Reviewer 2:

Comments: In this article, Su et al. reported a flavonoids natural product, i.e. myricetin, as a SARS-CoV-2 3CLpro/Mpro covalent inhibitor with pyrogallol as a novel electrophile warhead attacking the thiol group of the catalytic Cys145. The authors first screened a panel of flavonoids compounds against Mpro and identified two hits, including myricetin and dihydromyricetin. Following the measurement of low-micromolar range IC50 and antiviral EC50 values for Myricetin and its derivatives, co-crystal structures were further determined for myricetin, comp.3 and 7 at high resolutions (better than 2.5Å). Based on the electron density maps and mass spec results, the authors discovered the covalent mechanism of action for these myricetin derivatives. Surprisingly, Cys145 of Mpro covalently conjugated to the 2'-carbon of myricetin. Therefore, the authors employed quantum chemistry calculation to understand the mechanism of this reaction. In addition, the authors also worked on phosphonate derivatives of myricetin as prodrugs in order to improve their PK and PD properties.

Response: We thank the reviewer for the positive comments.

Comments: Since the outbreak of COVID-19, quite a few covalent inhibitors have been reported for SARS-CoV-2 Mpro so far. The authors previously published several Mpro inhibitor papers (Jin et al., Nature, 2020 and Dai et al., Science, 2020). In comparison with their own papers in Nature and Science, these myricetin derivatives are all at the same pocket of Mpro and share a covalent nature. Therefore, these myricetin derivatives are not new in terms of mechanisms. In comparison with another flavonoid non-covalent inhibitor Baicalein, reported by the same authors (ref 24), myricetin takes the same binding site but with a covalent mechanism. Because some covalent inhibitors have shown superior potency, e.g. GC376 ($k_i=60$ nM, Ma et al., Cell Research, 2020), myricetin and its derivatives are not best-in-class inhibitors either.

However, as the authors mentioned, this article might be the first reporting pyrogallol as a novel electrophile warhead attacking the catalytic Cys of any Cysteine proteases. The mechanism for the conjugation of Cys145 SH to the 2' carbon of pyrogallol is novel. Given the severity of the global COVID-19, this novel natural product covalent inhibitor for SARS-CoV-2 Mpro with an unexpected mechanism can potentially be of interest to the readers of Nature Communications.

Response: We thank the reviewer for recognition of our previous work and also agree with the reviewer on these comments. The novelty as well as the highlight of the present work is discovery of the unexpected mechanism of pyrogallol reacting with the catalytic cysteine of 3CL protease for the first time. We are grateful for the comment that our work can potentially be of interest to the readers of *Nature Communications*.

Comments: To better understand the intrinsic reactivity and gain insights into the mechanism, can the authors provide in vitro reactivity assessments of inhibitors with some sulfur nucleophiles such as GSH or N-Acetyl-L-cysteine (simple model compound to mimic cysteines in proteins) as suggested in

literatures (J. Med. Chem. 2014, 57, 10072–10079; Med. Chem. Commun., 2016, 7, 864–872)?

Response: Thank the reviewer for this suggestion. Indeed, we measured the in vitro reactivity of myricetin with GSH referring to the literature (J. Med. Chem. 2014, 57, 10072–10079) recommended by the reviewer with some modification, and the results were shown in Fig. 3c, d in the original version of the manuscript. As myricetin has strong UV absorption at 360 and 260 nm, we quantified the remaining myricetin with ultra-high-performance liquid chromatography previously. Nevertheless, the literature (J. Med. Chem. 2014, 57, 10072–10079) also suggested that MS or NMR method could detect the existence of the covalent adducts and trap the intermediates in the reaction. Therefore, following the reviewer's suggestion, we re-measured in vitro reactivity of myricetin (or N-phenylacetylamine, a positive control) with GSH using LC/MS as described in the literature (J. Med. Chem. 2014, 57, 10072–10079) and the resulting data are now shown in Fig. 3e, f in the revised manuscript and Supplementary Fig. 3-4 in the revised supplementary information. The concentration of myricetin and myricetin-GSH covalent adduct was detected. It shows that myricetin decreased whereas the adduct increased with the increase of reaction time (Supplementary Fig. 3). The resulting half time for myricetin reacting with GSH ($\text{GSH}t_{1/2}$) is 497.46 ± 29.54 min, close to the value (398.25 ± 10.93 min) obtained by using ultra-high-performance liquid chromatography. As a positive control, the half time for N-phenylacetylamine reacting with GSH is 33.90 ± 1.43 min (Lines 240-241, Page14).

Comments: Can the authors use MS or NMR or another approach to demonstrate the existence of the covalent adducts or trapped reaction intermediates in the reaction model proposed in Fig. 4c?

Response: According to the reviewer's suggestion, we used NMR and LC/MS to analyze the reaction solution. The covalent adduct has been traced by NMR and MS. The spectrum for this covalent adduct were added in revised

supplementary information (Pages 54-56) in the revised supplementary information. However, we didn't trap the reaction intermediate probably due to the trace amount of the intermediate(s).

Comments: Flavonoids are promiscuous. The authors investigated the selectivity of myricetin over cysteine proteases. How about over other proteins or enzymes? There are numerous PDB structures for flavonoids and/or myricetin bound kinases, e.g. PDB 6M88 and 2O63.

Response: We agreed that some flavonoids including myricetin might bind to other proteins such as the kinases mentioned by the reviewer. In these two kinases (PDB codes: 6M88 and 2O63), myricetin utilizes a noncovalent binding mode, while it covalently reacts with 3CLpro in our case. Given that the promiscuous binding of myricetin could result from noncovalent and covalent mode of action, it is hard to fully assess the selectivity of myricetin over all proteins. As the reviewer mentioned before that the significance of our work is to identify the pyrogallol group as a completely new warhead that could covalently linked to cysteine, we focused on the selectivity of the pyrogallol of myricetin reacting with other cysteine or serine proteases. In addition, myricetin actually serves as a good lead compound for further optimization which will improve the selectivity as well. For example, we docked myricetin and compound **3** into one kinase structure (PDB code: 2O63). The result is shown below. Although the difference between myricetin and compound **3** is only the methylation of the 7-OH group, the binding energy revealed by the docking score of compound **3** is much lower than that of myricetin (-4.699 vs -8.522; Figure R1). The methylation of the 7-OH in compound **3** breaks a strong hydrogen-bond between the 7-OH of myricetin and D168, leading to the weaker binding of **3** with the kinase. The structural modification on myricetin could thus reduce the promiscuous binding property of myricetin. Nevertheless, in further development with the structural modifications on myricetin and its derivatives, much attention should be paid to the selectivity study of new inhibitors.

Figure R1. The alignment between crystal structure of Pim1 in complex with myricetin (grey) and docked compound 3 (orange). The complex crystal structure of Pim1 (PDB code: 2O63) with myricetin was prepared using the Protein Preparation Wizard implemented in the Schrödinger suite. The respective docking grids were centered on the centroid of ligands using Grid Preparation tools. The 3D structures of myricetin and compound 3 were generated and optimized using the Ligprep tool of the Schrödinger suite, and the docking of compounds to Pim1 was performed with standard precision calculations with default settings. The final pose of every docking was selected from the top-scoring conformations.

Comments: How important is HIS41 in the covalent bond formation between myricetin and C145? Can the authors show experimentally what will happen if HIS41 is mutated, in terms of myricetin inhibitory activity and covalent adduct formation with Mpro.

Response: We thank the reviewer for this constructive suggestion. In order to evaluate how important His41 in the covalent bond formation, we mutated His41 to Ala41 so that it could not accept or transfer a proton during the reaction. Such a mutation (H41A) leads to completely inactive of the SARS-CoV-2 3CLpro in the FRET-based enzymatic assay, demonstrating the important role of His41 in hydrolysis of the substrate by the protease. As shown in Figure R2, SARS-CoV-2 3CLpro(H41A) did not show any enzymatic activity even at a high concentration of 10 μ M. In contrast, wild-type SARS-COV-2 3CLpro shows a

good enzymatic activity at a concentration of 100 nM. Therefore, we could not measure the inhibitory activity of myricetin on the SARS-CoV-2 3CLpro (H41A).

Figure R2. The enzymatic activity measurement of the wide-type SARS-CoV-2 3CLpro and the SARS-CoV-2 3CLpro(H41A) in hydrolysis of the fluorescently labelled substrate, MCA-AVLQSGFR-Lys(Dnp)-Lys-NH₂.

In addition, to address if H41A mutation has an influence on the binding of myricetin to the SARS-CoV-2 3CLpro, we employed the native MS to assess the ratio of the wild-type or the variant (H41A) bound with myricetin under the same condition. The wild-type and H41A mutation of SARS-CoV-2 3CLpro (1 μM) were incubated with 5 μM myricetin at 4°C for 3 h, respectively. Then, the samples were analyzed with a scimaX magnetic resonance mass spectrometry (MRMS) system (Bruker Daltonik GmbH, Bremen, Germany). It shows that the ratios of the wild-type and H41A mutation bound with myricetin were 76.4% and 34.5%, respectively (Figure R3). This suggests that His41 is also important for the binding of myricetin with the SARS-CoV-2 3CLpro.

Figure R3. Native-state mass spectrometry analysis of the wild-type (left) or the H41A variant (right) of SARS-CoV-2 3CLpro treated with methanol or myricetin.

Furthermore, to investigate if myricetin covalently binds to the H41A variant, we denatured the variant with formic acid after incubation for 5 h and analyzed the sample with MS. It shows that myricetin could covalently bound with the H41A variant (Figure R4). However, the ratio of the H41A variant covalently bound with myricetin was significantly less than that of the wide-type covalently bound with myricetin (Figure R4). This result further shows that H41A is crucial for the covalent reaction of myricetin with the SARS-CoV-2 3CLpro.

Figure R4. Mass spectrometry analysis for the wild-type and the H41A mutation of SARS-CoV-2 3CLpro treated with myricetin, and subsequently denatured by formic acid.

Comments: Line 661, “at least two independent experiments were carried out for each compound”, would duplicates for EC₅₀ measurement be acceptable for nature communications?

Response: As the same response for the previous comment raised by reviewer 1, we have carried out more independent experiments to determine the EC₅₀ value of each compound (Lines 675-678, Page 36). A total of six independent experiments (each experiment in triplicate) were performed for myricetin, dihydromyricetin, **3**, **7**, **9**, and **10** which are presented in the main text. Three independent experiments (each experiment in triplicate) were performed for compounds **4**, **5**, **6**, and **8** shown in supplementary information.

Comments: Line 48, “three global pandemics”, SARS and COVID-19 have been declared by WHO as pandemic. Is MERS too?

Response: We have changed the “three global pandemics” into “three epidemics” in the revised manuscript (Line 49, Page 4).

Comments: Line 81, ref 12 is a research paper about neutralizing antibody for the spike protein of SARS-CoV-2, did it show that Mpro is one of the best targets for CoVs? I would suggest the authors double check.

Response: We are so sorry for this misquotation. The right reference has been cited in the revised manuscript (ref 12).

Comments: Ref 13 and ref 31 are the same paper?

Response: They are the same paper. We have deleted the redundant ref 31 in the revised manuscript. We thank the reviewer for thoroughly reading our paper. We have carefully checked all references as well as their citations in the revised manuscript.

Comments: Line 139, PCR was used for EC50 measurement, PCR for the expression of what protein? What primers have been used?

Response: PCR were conducted for measurement of the copy numbers of the receptor binding domain (RBD) of the SARS-CoV-2 spike protein. The primers used for quantitative PCR were RBD-qF1: 5'-CAATGGTTTAAACAGGCACAGG-3' and RBD-qR1:5'-CTCAAGTGTCTGTGGATCACG-3. We have added these information to the method in the revised manuscript (Lines 672-675, Page 35).

Comments: Line 358, IC50 increase from 0.31 to 0.67 uM, is this a one-fold increase or two?

Response: It is two-fold increase. It has been corrected in the revised manuscript (Lines 360, Page 21).

Comments: Line 398, IC50s of comp 7 and 3 are 0.26 and 0.31 uM respectively (based on Suppl Table 2), these two are essentially the same, no one is better than the other one. Why it was stated “even better than”?

Response: We agreed with the reviewer’s comment and revised the inappropriate description. We have changed “even better than” into “comparable to” in the revised manuscript (Line 402, Page 23).

Comments: Line 691, -80 XXX?

Response: it is -80°C. This error was made when converting word file to pdf file. It has been corrected in the revised manuscript (Line 712, Page 37).

Comments: Line 708, PDB 6M2N was used in the calculation, 6M2N is the baicalein bound Mpro. why not the myricetin-bound structure used in the calculation, e.g. 7DPP or 7DPU?

Response: The structure used in the calculation is indeed an apo structure of the SARS-CoV-2 3CLpro. We apologized for providing the wrong PDB code in the original version. The correct PDB code is 6M2Q (apo SARS-CoV-2 3CLpro). We have corrected it in the revised manuscript (Line 731, Page 38). The apo structure of the dimeric protease was used in order to study the conformational changes of the protein or residues without the interference of the ligand binding.

Comments: In supplementary materials, suppl figures 9 and 10 were missing in table of contents , line 24.

Response: Supplementary figures 9-11 have been added in table of contents in the revised supplementary information.

Comments:

Line 271, was added to labeled  was added to label

Line 328, owe to  owing to

Line 387, are not occurred  did not occur

Line 515, new warhead that could covalent modification of

Response: Again, we sincerely thank the reviewer for thoroughly reading our manuscript. we have corrected these grammar mistakes (Line 274, Page 17; Line 329, Page 19; Line 390, Page 22; and Line 522, Page 28). In addition, a thorough proofreading and grammar check have been done.

Responses to Reviewer 3:

Comments: The manuscript “Identification of pyrogallol as a novel warhead in the design of covalent inhibitors for the SARS-CoV-2 3CL protease” reports the identification of myricetin and derivatives as covalent inhibitors of SARS-CoV-2 3CL protease. This study includes experimental characterization of compound potency, selectivity, determination of binding mode by crystallography, and antiviral activity. Computational studies (molecular dynamics simulations and quantum chemistry calculations) have also been performed to study the covalent reaction mechanism and energies involved. There is a clear contribution of the study in terms of characterizing a new class of SARS-CoV-2 3CL protease inhibitors, in a fairly complete study. Thus, this study could potentially be appropriate for publication in Nature Communications. However, I recommend improvement of the following points before a new evaluation of this manuscript.

Response: We thank the reviewer for the positive comments and finding our work potentially appropriate for publication in *Nature Communications*.

Comments: Figure 4 – The schematic panel in Figure 4c is not very informative. Since the conformation of the catalytic His is discussed as important to define whether Cys145 reacts with each compound, it would be better to have tridimensional figures clearly illustrating the different conformations of His41 in these simulations.

Response: Thank the reviewer for this constructive suggestion. The Fig. 4 have been modified following the reviewer’s suggestion. The tridimensional

figure showing the different conformations of His41 have been added to Fig. 4 in the revised manuscript (Page 20).

Comments: Lines 388-392 – Based on Supplementary Figure 9, in my opinion, this is not a convincing assertion. There is indeed a rotation in the ring, but it does not constitute “a surprisingly different” binding mode. I also disagree with emphasizing that potency is improved, based on only a two-fold difference in IC50 values. This is a minor potency improvement.

Response: We have deleted the “surprisingly” and “with improved potency” in the revised manuscript (Line 396, Page 23). The modified sentence is “The derivation of compound 3 from myricetin presents an example in which a minor chemical modification on the compound leads to a different binding pose.”

Comments: Lines 461-467. Please cite references that support the discussion that the PK profile for compound 7 is acceptable.

Response: A compound with oral bioavailability above 10% has a potential for development into an oral drug. The oral bioavailability of compound 7 is 18.1%, it is thus acceptable for the potential of development into an oral drug. Following the reviewer’s suggestion, the related reference has been cited in the revised manuscript (refs 15 and 34).

Comments: Lines 474-475. Please clarify why Cys is considered an “intriguing amino acid for the design of targeted covalent inhibitors”. To my knowledge, Cys is recognized as one of the best nucleophiles and it is likely the most exploited residue for the design of covalently bound inhibitors.

Response: We agree with the reviewer on this point and modified the description into “The catalytic cysteine of 3CLpro presents one of the best nucleophiles for the design of covalently bound inhibitors.” (Lines 483-484, Page 27).

Comments: The current conclusion is very long and contains much text which does not refer to the conclusions of this manuscript. Some content is also repetitive, as in the description of pyrogallol as a novel warhead.

Response: We have changed the “conclusion” section into “discussion” according to the requirements of *Nature communications*. Following the reviewer’s suggestion, the contents such as the description of pyrogallol as a novel warhead and the comparison between baicalein and myricetin have been deleted to condense the discussion (Lines 527-555, Pages 29-30).

Comments: Lines 483-485. Dozens of structures of SARS-CoV-2 Mpro bound to covalent inhibitors are available. Thus, it cannot be said that this manuscript is the first report of small molecule covalent inhibitors of this enzyme.

Response: We have modified the description and changed “the first” to “a non-peptidomimetic” (Line 493, Page 27). The revised sentence is “Herein, we describe the state-of-the-art of the cysteine-directed chemical modification by the natural product and report a non-peptidomimetic covalent inhibitor of 3CLpros.”

Comments: Lines 494-497 – Can the antiviral activity of compound 9 be considered “excellent”?

Response: We used the word “improved” instead of “excellent” in the revised manuscript (Line 454, Page 25).

Comments: And is the PK profile of compound 7 “greatly improved”? Based on the current manuscript and data provided, these assertions from the authors are not clear to me.

Response: The word “greatly” was deleted in the revised manuscript (Line 504, Page 28).

Comments: Lines 500-503. This is not the first report of antiviral activity for myricetin and derivatives. Please see publications such as: DOI: 10.1080/14756366.2020.1754813, DOI: 10.1016/j.antiviral.2020.104714.

Response: We modified this description and cited the references (Lines 508-512, Page 28).

Comments: In the methods session, please add more information on the following topics: What was the minimum purity of the compounds synthesized?

Response: We have added the sentence “The purity of the synthesized compounds is over 95%” into the revised manuscript (Lines 560-563, Page 30). In addition, the purity of each compound and the spectrum of HPLC, NMR and MS for all compounds have been added (Pages 32-56 in revised supplementary information).

Comments: Please provide the gene accession code that allows the reader to obtain the exact sequence of the Mpro and PLpro expressed.

Response: The genbank number of the Mpro and PLpro expressed, have been added (Lines 571-572 and 586, Page 31). The genbank number for SARS-CoV-2 Mpro and PLpro is MN908947.3. The genbank number for SARS-CoV Mpro is AAP13442.1.

Comments: Please indicate the maximum amount of DMSO used in the assays and inform what is the impact of such DMSO concentration in the activity of the enzymes.

Response: The final concentration of DMSO in the enzymatic assays is less than 2% of the total volume, which had no effect on the enzyme activity of the SARS-CoV-2 3CLpro, SARS-CoV 3CLpro, SARS-CoV-2 PLpro, and chymotrypsin (Lines 632-635, Page 34). The final concentration of DMSO in the cell-based antiviral activity assay and Gel-Based Competitive ABPP Assay is less than 0.1% and 0.5% of the total volume, respectively (Lines 668-669,

page 35 and Line 718, Page 37). These descriptions have been added into the revised manuscript.

Comments: Please report the minimum number of compound concentrations for each IC₅₀ curve and the number of replicates in each independent experiment.

Response: For each compound, three independent experiments and each independent experiment in duplicate were performed for the determination of the IC₅₀ values (Lines 630-631, Page 33). To obtain EC₅₀ values of the compound inhibiting the replication of SARS-CoV-2, six independent experiments (each experiment in triplicate) were performed for myricetin, dihydromyricetin, **3**, **7**, **9**, and **10**, and three independent experiments (each experiment in triplicate) were performed for **4**, **5**, **6**, and **8** (Lines 675-678, Page 36). These descriptions have been added in the revised manuscript.

Comments: In the molecular dynamics protocol, please mention which protonation states were predicted for the titratable active site residues, when using H++.

Response: H++ (<http://biophysics.cs.vt.edu/H++>) is a free open-source web server for calculating the pKs of titratable groups including the side chains of Asp, Glu, Arg, Lys, Tyr, His or Cys and thus predicting their protonation states (Nucleic Acids Res., 2012, 40:537-541). In this study, the protonation states of all titratable residues were evaluated at physiological pH (7.4), giving the deprotonation states of Asp and Glu and protonation states of Arg, Lys, Tyr, and Cys throughout the 3CLpro dimer structure (consistent with the standard AMBER protonation states at physiological pH). In addition, the His residues adopt the neutral deprotonation states but display different hydrogen orientations, e.g., proton presents at the HD1 position of His164 but at the HE2 position in His41, His163 and His172 in the active site. These information has been added in the revised manuscript (Lines 732-736, Page 38).

Comments: Supplementary Tables 4-7 provide details on calculated energies from quantum mechanical calculations. However, these tables are not discussed anywhere in the manuscript. It is important to discuss these results, at least in the supplementary material.

Response: We agree with the reviewer that Supplementary Tables 5-8 (Supplementary Tables 4-7 in original manuscript) give details on calculated energies from the QM calculations in which the Gibbs free energy of each substance is obtained by adding the thermal correction with the single-point energy, and the imaginary frequency (IF) is used to evaluate whether a substance is at a local energy minimum or a transition state. With these information, we achieved the relative Gibbs free energy of the transition state and product with respect to the reactant (by subtracting the free energies of the substances involved in the transition state or product to those in the reactant) and thus draw the reaction pathway like Fig. 4b. Following the reviewer's comment, we rewrote the "Ab Initio Calculation" (Pages 40-41) subsection to emphasize how the Gibbs free energy is obtained based on the detailed information included in Supplementary Tables 5-8.

Comments: Some additional minor points for improvement: Lines 432-434 - The difference between the original hits and compounds 8 and 9 is not just phosphorylation. Please clarify the text to give more precise information about the substituent added to these derivatives.

Response: The detailed description of the added groups has been provided in the revised manuscript (Lines 440-441, Page 25). The sentence is revised to "Accordingly, a proof-of-concept prodrugs (compounds 8 and 9) were obtained by adding two kinds of phosphate groups (5,5-dimethyl-1,3,2-dioxayl phosphate and diphenyl phosphate) to the 7-OH of myricetin (Supplementary Fig. 1a and Supplementary Table 2) in order to improve the aqueous solubility as well as the membrane permeability of myricetin."

Comments: In Figures 2, the color scheme with carbons in magenta makes it very difficult to distinguish Carbons from Oxygens (red). Please employ a color scheme with better contrast between Carbon and Oxygen.

Response: The carbons color of compound 7 in figure 2c has been changed into pink (Page 12).

Comments: In Supplementary Figure S1, the PDB codes listed do not correspond to all available SARS-CoV-2 Mpro structures. Please modify the figure legend or add other structures to the figure. Also, highlight/clarify which structure is bound to baicalein, since the manuscript references this figure when discussing binding to this compound.

Response: We have re-superposed a total of 306 structures of the SARS-CoV-2 3CLpro available in Protein Data Bank on April 1, 2021 with the complex structure of SARS-CoV-2 3CLpro/myricetin (Supplementary Fig. 1b in the revised supplementary information). The figure legend has also been modified. The structures bound with baicalein and myricetin were highlighted by sticks.

With these changes we hope we have addressed all comments. We would like to thank the reviewers again for their constructive, professional and helpful suggestions.

REVIEWER COMMENTS

Reviewer #2 (Remarks to the Author):

The authors have made great efforts to improve their manuscript and they have addressed all my comments.

I am impressed with their new experiments showing if the covalent binding of myricetin to Mpro is affected by the H41A mutation (Figure R3 and R4). I would suggest them to add these results to the manuscript or to the supplementary material, but I would leave this up to the authors to decide.